

# CORDEX-WRF v1.3: Development of a module for the Weather Research and Forecasting (WRF) model to support the CORDEX community

Lluís Fita[1], Jan Polcher[2], Theodore M. Giannaros[3], Torge Lorenz[4], Josipa Milovac[5], Giannis. Sofiadis[6], Eleni Katragkou[6], and Sophie Bastin[7]

[1]Centro de Investigaciones del Mar y la Atmósfera (CIMA), CONICET-UBA, CNRS UMI-IFAECI, C. A. Buenos Aires, Argentina
[2]Laboratoire de Météorologie Dynamique (LMD), IPSL, CNRS, École Polytechnique, Palaisseau, France
[3]National Observatory of Athens (NOA) - Institute for Environmental Research and Sustainable Development (IERSD), Penteli, Greece
[4]Uni Research Climate, Bjerknes Centre for Climate Research, Bergen, Norway
[5]Institute of Physics and Meteorology, University of Hohenheim, Stuttgart, Germany
[6]Department of Meteorology and Climatology, School of Geology, Aristotle University of Thessaloniki (AUTH), Thessaloniki, Greece
[7]Laboratoire Atmosphères, Milieux, Observations Spatiales (LATMOS)/IPSL, UVSQ Université Paris-Saclay, Sorbonne Université, CNRS, Guyancourt, France

**Correspondence:** L. Fita (lluis.fita@cima.fcen.uba.ar)

**Abstract.** The 'Coordinated Regional Climate Downscaling Experiment' (CORDEX) is a scientific effort of the World Climate Research Program (WRCP) for the coordination of regional climate initiatives. In order to accept an experiment, CORDEX provides experiment guidelines, specifications of regional domains and data access/archiving. CORDEX experiments are important to study climate at the regional scale, and at the same time, they also have a very prominent goal in providing regional

climate data of high quality. Data requirements are intended to cover all the possible needs of stake holders, and scientists working on climate change mitigation and adaptation policies in various scientific communities. The required data and diagnostics are grouped into different levels of frequency, priority, and some of them even have to be provided as statistics (minimum, maximum, mean) over different time periods. Most commonly, scientists need to post-process the raw output of regional climate models, since the latter was not originally designed to meet the specific CORDEX data requirements. This post-processing

procedure includes the computation of diagnostics, statistics, and final homogenization of the data, which is often computationally costly and time consuming. Therefore, the development of specialized software and/or code is required. The current paper presents the development of a specialized module (version 1.3) for the Weather Research and Forecasting (WRF) model, capable of outputting the required CORDEX variables. Additional diagnostic variables not required by CORDEX, but of potential interest to the regional climate modeling community, are also included in the module. 'Generic' definitions of variables

are adopted in order to overcome model and/or physics parameterization dependence of certain diagnostics and variables, thus facilitating a robust comparison among simulations. The module is computationally optimized, and the output is divided in different priority levels following CORDEX specifications (Core, Tier1, and additional) by selecting pre-compilation flags.



This implementation of the module does not add a significant extra cost when running the model, for example the addition of the Core variables slows the model time-step by less than a 5%. The use of the module reduces the requirements of disk storage by about a 50%.

# 1 Introduction

Regional climate downscaling pursues the use of limited area models (LAM) to perform climate studies and analysis (Giorgi and Mearns, 1991). It is based on the premise that, by using LAM, modelers can simulate the climate over a region at a higher resolution as compared to the Global Climate Models (GCM). Therefore, certain aspects of the climate system can be better represented due to the higher resolution and higher complexity of parameterizations (inherent of the LAM models) used to simulate physical processes which can not be explicitly resolved (e.g.: short/long-wave radiation, turbulence, dynamics of water species). This methodology has been widely used both for studying climate features, connections and processes (Jaeger and Seneviratne, 2011; Knist et al., 2014; Kotlarski et al., 2017), and to produce climate data within the scope of continental, national or regional climate change studies.

The 'Coordinated Regional Climate Downscaling Experiment' (CORDEX, http://www.cordex.org/) of the World Climate Research Program (WRCP) aims to organize different initiatives devoted to regional climate all around the globe following a similar experimental design (Giorgi et al., 2009; Giorgi and Gutowski, 2015). CORDEX, with the second phase being currently under discussion, attempts to establish a series of criteria for dynamical downscaling experiments, which include setting of common domain specifications and horizontal resolutions in order to make sure that all the continental areas of the Earth are under study (e.g. in 2010 Africa was a priority and researchers worldwide volunteered to contribute with their own simulations). Furthermore, CORDEX sets a series of model configurations (e.g. GCM forcing, Greenhouse Gas (GHG) evolution) to ensure that model simulations are carried out under similar conditions and therefore are inter-comparable. At the same time, CORDEX requires a list of variables necessary for a later use of model data for multi-models analysis and other climate-related research activities like climate change mitigation, adaptation and stake holders decision making policies. In order to maximize and facilitate data access, (mostly made available by the Earth System Grid Federation (ESGF), https://esgf.llnl.gov/), these data have to be provided also following a series of homogenization criteria known as Climate and Forecast (CF) compliant (http://cfconventions.org/) which comes from the Coupled Model Intercomparison Project (CMIP) exercises. The list of variables required by CORDEX consists of standard model fields and some diagnostics in certain frequencies, and statistical aggregations such as minimum, maximum or mean for a given period. These variables are grouped into different priority levels ('Core', 'Tier1' and 'Tier2'), with 'Core' being the mandatory list of variables.

The production of these data sets is not a simple task and usually represents a big issue for the modelling community. Regional climate experiments tend to produce large amounts of data, since scientists simulate long time periods at high resolutions. Modelers have to code a software at least capable of: (1) computing a series of diagnostics, (2) concatenating model output, (3) performing statistical temporal computations and (4) producing data following CF-compliant (i.e. cmorization)





criteria in netcdf format. Apart from being time-consuming due to its complexity and the process management, this codification also implies certain duplication of huge data-sets and additional consumption of computational resources.

Several tools (e.g. NetCdf Operators - NCO, Climate Data Operators - CDO) exist that allow to easily manipulate netcdf files (extract/concatenate/average/join etc.), and also some other post-processing initiatives that have been available especially to the Weather Research and Forecasting (WRF; http://www.mmm.ucar.edu/wrf/users/; Skamarock et al., 2008) community: WRF NetCDF Extract&Join (wrfncxnj, http://www.meteo.unican.es/wiki/cordexwrf/SoftwareTools/WrfncXnj), wrfout_to_cf.ncl (http://foehn.colorado.edu/wrfout_to_cf/), METtools (https://dtcenter.org/met/users/metoverview/index.php), and Climate Model Output Rewriter (CMOR, https://cmor.llnl.gov/).

WRF is a popular model for regional climate downscaling experiments. It is used world-wide in different CORDEX domains (Fu et al., 2005; Mearns et al., 2009; Nikulin et al., 2012; Domínguez et al., 2013; Vautard et al., 2013; Evans et al., 2014; Katragkou et al., 2015; Ruti et al., 2016). The model was initially designed for short-term simulations at high resolutions, but a series of modifications that had been introduced to the model code so far enhanced its capabilities, and made it appropriate also for climate experiments (Fita et al., 2010). Since WRF does not directly provide most of the required variables for CORDEX and due to the complexity of the post-processing procedures, many of the already existing WRF-climate simulations are not publicly available to the community.

We present a series of modifications to the model code, and a new module (version 1.3) which will enable climate researchers using WRF to get almost all the CORDEX variables directly in the model output. With the use of this module, production of the data for regional climate purposes will become easier and faster. These modifications directly provide the required fields and variables ('Core' and almost all 'Tier1') during model integration, and aim to avoid the post-processing of the WRF output up to certain level. However, in this version, they do not cover all the previously mentioned aspects of the task, such as the cmorization of the data. The data cmorization can be defined as a series of processes that need to be applied to the model output in order to meet the standards provided under the CF guidelines (which follows the C-MOR standard, https://pcmdi.github.io/cmor-site/). These guidelines are designed to facilitate the comparison between climate models, and they represent the standard for the 'Coupled Model Intercomparison Project' (CMIP, https://cmip.llnl.gov/). This standardization includes the file names, variable names and metadata (units, standard names and long names), specification of geographical projections and time axis. In order to achieve the standardization of WRF output to completely follow CF requirements, it would be necessary to change WRF input/output (I/O) tools which would affect backward compatibility. Therefore, the users of the CORDEX-WRF module will still need to perform part of the standardization by themselves. This includes joining/concatenating of WRF files, performing additional temporal statistics (e.g. daily mean, monthly minimum), using standard names and attributes of the variables, file names, and finally providing the right variable with the standard attributes to describe the time coordinate.

The modifications also aim to establish a series of homogenization of certain diagnostics. These diagnostics can be computed following different methodologies, and consequently they may be model and/or even physical parameterization dependent. In order to avoid dependency on the model configuration (mainly sensitivity to the choice of the available different physical schemes), and to allow for a fair comparison between different simulations, a series of additional ´generic' definitions of some diagnostics are presented when possible.





The modification of the WRF model code was initiated within the development of the regional climate simulation platform from the Institute Pierre Simone Laplace (IPSL) - RegIPSL (https://sourcesup.renater.fr/wiki/morcemed/Home) and the CORDEX Flagship Pilot Study (CORDEX-FPS), *'Europe+Mediterranean; Convective phenomena at high resolution over Europe and the Mediterranean'*, (Coppola et al., 2018) in order to obtain the variables required for the CORDEX experiment (available at: https://www.hymex.org/cordexfps-convection/wiki/doku.php?id=protocol) and share the code among the WRF users of the CORDEX-FPS experiment.

In this work the complete module is presented, its capabilities are demonstrated, and the results of several diagnostics are shown in order to illustrate the accuracy of the implementation. The initial section of the paper describes the modifications that have been introduced into the code followed by a description of the variables required by CORDEX. The following section demonstrates the performance tests, and gives a description of aspects which are currently missing, but necessary to be added. The paper finishes with a discussion and outlook section.

## 2 The CORDEX module

Here we introduce the module and we explain the modifications introduced in the model. The steps necessary to follow in order to compile and use the module are provided as well. The module has been implemented following standards of modularity which facilitates the upgrading and the introduction of new variables to it. Apart from the modifications of the code of the WRF model, the complete module currently consists of two modules:

- `phys/module_diag_cordex.F`: Main module which manages the calls to the variables and performs the necessary accumulations for the calculations of statistical values (e.g.: mean, maximum, minimum)

- `phys/module_diagvar_cordex.F`: Module which contains the calculations of all the CORDEX variables separated in individual and independent 1D Fortran subroutines.

The structure of the variables in the WRF code is managed throughout a series of ASCII files called `registry`. At the same time, WRF model set-up is managed though the use of a Fortran `namelist` statement which reads the ASCII file called `namelist.input` which has different sections. Following WRF management, the module is accompanied with a new registry file called `Registry/registry.cordex` where the variables, and namelist parameters related to the module are defined. The specific set-up of the module is managed in the WRF namelist in a new section called `cordex`. Additional necessary modifications done in different WRF modules are: (1) the call to the main module has been added to `phys/module_diagnostics_driver.F`, which is a module that accounts for the management of diagnostics, (2) in the general `Registry/Registry.EM_COMMON` an input to the `registry.cordex` has been added, (3)the complementary interpolated variables have been introduced in the registry file for the pressure interpolated variables `Registry/registry.diags`, (4) in the module which computes the pressure interpolation `phys/module_diag_pld.F` the complementary interpolated variables have been added, and (5) in `dyn_em/start_em.F` the initialization of the modified pressure interpolation has been added. Furthermore, some modifications have been done in the `main/depend.common` and `phys/Makefile` to





get the new module compiled. For the inclusion of the water-budget variables some specific changes have been introduced in the `dyn_em/solve_em.F` module in order to get the advection terms of all water species. Finally, a descriptive file called `README.cordex` with the description and synthesized instructions for compilation and use is provided as well.

Apart from the directly added variables, WRF model (afte version V3.4.1) can provide 3D dimensional fields either on pres-
5 sure or height levels. However, CORDEX protocol requires additional variables which have been added to this specific output by introducing some modifications into `Registry/registry.diags`, `phys/module_diag_pld.F`, `dyn_em/start_em.F` and `phys/module_diagnostics_driver.F`.

Output of the module is grouped in a single file (WRF's auxiliary history output #9) with a file name: `wrfcordex_d<domain>_<date>` with the standard WRF parameters of: output frequency, number of time steps per file
and format. Additional CORDEX variables required at pressure levels have been included in the WRF auxiliary output file number 23 which provides certain variables interpolated at pressure levels. These introduced CORDEX variables follow the file set-up via the currently existing namelist section called `diags&`.

## 2.1 Module use

A series of steps have to be made in order to use the CORDEX-WRF modifications. These steps encompass compilation of the
15 module and its specific set-up to be used during the execution time of the model and are described in the following subsections. A more detailed explanation is available on the wiki page of the module: http://wiki.cima.fcen.uba.ar/mediawiki/index.php/CDXWRF

### 2.1.1 Compilation

Compilation of the module requires to set up different pre-compilation flags according to the needs of the user. It is necessary to
20 keep in mind that this is done due to efficiency constrains (see below in section 6 ´Optimization'), although it is not a common procedure in the standard use of WRF. Usually WRF has almost all options available from a single compilation switching options via the namelist.

With the pre-compilation flag `CORDEXDIAG`, the compilation of the parts of the module which provide 'Core' CORDEX variablesis activated. The 'Tier1' (`CDXWRF=1`) and 'additional' (`CDXWRF=2` which also includes 'Tier1') groups of variables
can be selected via the additional pre-compilation flag `CDXWRF`. The reader is referred to Appendix for more details about the groups of CORDEX variables associated to each option.

The WRF model defines variables via a series of ASCII files contained in the `Registry` folder. These files contain the characteristics of the variables during model execution, mainly: name of the variable during execution, rank and dimensions of the variable, assigned output file, name of the variable in the output file, description of the variable, and units. WRF model
keeps all the variables in a Fortran pointer derived type (called `grid`). In order to adapt this derived type to the pre-selected compilation, it is necessary also to modify the module's specific register file (`register.cordex`) according to the chosen value given to the additional pre-compilation `CDXWRF` flag (if used). This is done in a way to control the size of `grid` derived type which has a positive impact on the model performance (see below).





According to the value given to the pre-compilation `CDXWRF` flag, different amount of variables is written out to the `'wrfcdx'` output file:

– Using `CORDEXDIAG` and without `CDXWRF`: all the CORDEX 'Core' variables will be calculated

– CDXWRF=1: CORDEX 'Core' + 'Tier1' variables clgvi, clhvi, zmla, [cape/cin/zlfc/plfc/lidx]{min/max/mean}

– CDXWRF=2: as with CDXWRF=1, plus additional 3D variables at model $\eta$-level (ua, va, ws, ta, press, zg, hur, hus), 2D variables (tfog, fogvisblty{min/max/mean}, tds{min/max/mean}), and the water-budget variables (wbacdiabh, wbacpw, wbacpw[c/r/s/i/g/h], wbacf, wbacf[c/r/s/i/g/h], wbacz, wbacz[c/r/s/i/g/h], wbacdiabh{l/m/h}, wbacpw{l/m/h}, wbacpw{l/m/h}[c/r/s/i/ wbacf{l/m/h}, wbacf{l/m/h}[c/r/s/i/g/h], wbacz{l/m/h}, wbacz{l/m/h}[c/r/s/i/g/h])

In the `'wrfcdx'` output a user in addition can get the instantaneous values of the CORDEX variables provided as statistics
(e.g. capemean, tdsmax, or all the water-budget variables). These extra variables can be retrieved following certain modifications (and re-compilation) in `phys/module_diag_cordex.F`, `phys/module_diagnostics_driver.F`), as well as in the registry file `registry.cordex`.

## 2.2 Usage

Modifications of the module include two main sets of variables: (1) new variables and diagnostics and (2) additional variables
interpolated at pressure levels. These two sets of variables are provided in two separated files. New auxiliary output file in the ninth stream provides all the new variables and diagnostics required by CORDEX. Additional pressure interpolated variables are included in the 23rd stream, which corresponds to the current one provided by WRF for these purposes. Each of these files have to be set-up in the namelist in the same way as it is done with the typical WRF output files. A new section labeled `cordex` has to be added into the WRF's namelist which allows to choose/set-up different options of the module. The description of all
the available options is provided in Table 1. In this section it is required to choose which implementation of certain diagnostics to use, to provide values to some parameters for certain diagnostics, and to activate/deactivate some of the most computationally costly diagnostics. Default values for all the options are provided in order to facilitate the use of the module.

This module has been tested under different High-Perfomance Computing (HPC) environments and compilations. It has been compiled with two compilers: gfortran and ifort. Different parallelization paradigms: serial, distributed memory and
25 hybrid (distributed and shared) and the parallelized version of the netcdf libraries. The tests have been performed mainly using 2-nested domains with the second one being at the convection permitting resolution (no cumulus scheme activated). Under all these circumstances the module worked as expected.

Since version v1.3 a text message with the version of the module is printed in the standard output at the first time-step of the model run in order to facilitate the detection of the module version that is being used.



**Table 1.** Set-up parameters of the `module_diag_cordex` module for the WRF namelist contained in the `cordex` section. See 'variables' sections for more details of the meaning of each methodology

| name & value | description | default value |
|---|---|---|
| output_cordex = 0 | CORDEX Diagnostic de-activation | 0 |
| output_cordex = 1 | CORDEX Diagnostic activation | 0 |
| psl_diag = 1 | sea-level pressure diagnostic following hydrostatic Shuell correction (Stackpole and Cooley, 1970) | 3 |
| psl_diag = 2 | psl diagnostic following a target pressure (Benjamin and Miller, 1990) | 3 |
| psl_diag = 3 | psl diagnostic following ECMWF method (Yesad, 2015) | 3 |
| psmooth = 5 | number of passes of neighbor filtering (mean of the grid point with its 8 neighbors) of psfc (only for psl_diag=2) | 5 |
| ptarget = 70000. | pressure [Pa] target to be used by psl_diag=2 | 70000. |
| wsgs_diag = 1 | wind-gust diagnostic following (Brasseur, 2001) | 1 |
| wsgs_diag = 2 | wind-gust following heavy precipitation method | 1 |
| output_wb = 0 | deactivation of the computation of water-budget variables (Fita and Flaounas, 2018) | 0 |
| output_wb = 1 | activation of the computation of water-budget variables | 0 |
| wsz100_diag = 1 | wind extrapolation at z100m_wind using power-law method | 1 |
| wsz100_diag = 2 | wind extrapolation at z100m_wind using logarithmic-law method | 1 |
| wsz100_diag = 3 | wind extrapolation at z100m_wind using Monin-Obukhov method | 1 |
| z100m_wind = 100. | height [m] to extrapolate winds for wsz100_diag | 100 |
| zmlagen_dqv = 0.1 | percentage of variation of mixing ratio to determine mixed layer depth used in zmlagen computation (Nielsen-Gammon et al., 2008) | 0.1 |
| zmlagen_dtheta = 1.5 | increment in K of potential temperature from its minimum within the MLD used in zmlagen computation | 1.5 |
| potevap_diag = 1 | potential evapotranspiration using bulk computation (Manabe, 1969) | 2 |
| potevap_diag = 2 | potential evapotranspiration using Milly92 correction (Milly, 1992) | 2 |
| convxtrm_diag = 0 | deactivation of diagnostic of extremes from convection indices | 0 |
| convxtrm_diag = 1 | activation of diagnostic of extremes from convection indices | 0 |
| fogvisibility_diag = 1 | diagnostic of visibility inside fog (Kunkel, 1984) | 3 |
| fogvisibility_diag = 2 | RUC method (Smirnova et al., 2000) | 3 |
| fogvisibility_diag = 3 | FRAML 50% prob (Gultepe and Milbrandt, 2010) | 3 |
| fogvars = 1 | use 3D variables (hur, closest level to surface) to diagnose fog | 1 |
| fogvars = 2 | use sfc variables (hurs) to diagnose fog (not available for fogvisibility_diag = 1) | 1 |



## 3 CORDEX variables

CORDEX requires a series of mandatory variables grouped in the 'Core' level, and additional variables grouped in 'Tier1' and 'Tier2' levels. Furthermore, CORDEX also requires to provide some statistics values from different variables. Some variables might be required to be provided at the same time as instantaneous value and as statistics in different groups. To meet the

5 CORDEX specifications, regional climate models have to provide three kind of variables:

- instantaneous: values obtained at each model integration time-step. An instantaneous value represents the field at the given instant of time all over the given space encompassed within the grid point.

- statistics: values obtained as a statistics of consecutive instantaneous values for a given period of time. The statistical computation could be: minimum, maximum, mean or accumulated value, as well as the flux. Thus, a statistics variable

represents the temporal statistics of the field for a given period of time all over the given space encompassed within the grid point. CORDEX guidelines also mainly requires different temporal aggregations: 3-hourly, daily, monthly and seasonal

- fixed: values which do not have an evolution in time. These fields are fixed all over the simulation.

The WRF I/O file managing system provides an infrastructure for more than 20 different output files at the same time. Each

15 file is independently managed, and therefore in the namelist a user has to set-up mainly 2 different options for each output stream: frequency of an output (frequency of writing out the variables to an output file in minutes, e.g. 30 , 60), number of frames per file (e.g. for 3-hourly frequency 8 frames per file will give a daily output). Variables that will be written out to each different stream (a variable can be written in multiple streams) can be selected within the previously mentioned 'registry' ASCII files. During the model integration, at the given time-step corresponding to the defined output frequency, data will be

written out to the output file. When a given file reaches the selected amount of frames, it is closed and a new one is open. The file name usually follows a criteria of a given header name (e.g. 'wrfcdx' for this module) and the current date of the simulation which is also set-up in the namelist.

The CORDEX-WRF module is designed to provide the variables which are not currently available in the standard WRF output using the 9th stream, without reducing any of the capabilities of the model. Following this criteria, the module uses the

25 same structure and components of the model designed to manage its I/O, and provides the CORDEX variables at the given 'Core' time-frequency. This means that the statistical values are directly provided using the internal values between output frequencies. This ensures that, for example, a minimum value would exactly be the minimum value that the model simulated between teh period of time. These variables are re-initialized after each stream output time (see figure 1). The WRF model is used in a myriad of applications and regions, thus it was decided that statistics values will be provided at the selected frequency

of the 9th stream. This gives more flexibility, allowing a user to get e.g. high frequency outputs. However, it will still be necessary to do the aggregation of the output files in order to provide the required CORDEX statistics at the 3-hourly, daily, monthly and/or seasonal periods. User is strongly encouraged to use the output frequencies for the 9th stream which are easy to combine in order to retrieve the required CORDEX statistics. It is necessary to highlight that the statistics for a given period



**Table 2.** Description of CORDEX additional pressure interpolated variables provided with the module

| CF name | WRF name | description | units |
|---------|----------|-------------|-------|
| hus | HUS_PL | specific humidity | 1 |
| wa | W_PL | vertical wind speed | m.s-1 |
| ua | UER_PL | Earth-rotated wind x-component | ms-1 |
| va | VER_PL | Earth-rotated wind y-component | ms-1 |
| ws | WS_PL | wind speed | ms-1 |

contained in the 9th stream corresponds just to the instant time of writing the field into the file (e.g. on a 3-hour frequency the value inside the file at [HH]:00:00 represents the statistics from [HH]:00:00 to [HH]+2:59:59). The WRF I/O does not allow to produce static/fixed fields, therefore this group of variables are not provided by the module.

### 3.1 Generic methodology

The list of variables requested by the CORDEX experiment (see the Table B in appendix) are intended to be useful for the climate change mitigation, adaptation and decision making communities. Note that for CORDEX-FPS some specific variables or different frequency of outputs can be added/removed to this list. Most of these variables are not directly available in WRF, thus we have to introduce them into the code of the model. Taking into account the performance of the model, variables are computed at specific frequencies: (1) at all time steps when a statistic value (accumulation/flux, minimum, maximum and/or

mean) of the variable is required, (2) at the given time step, when a variable that is used for the diagnostic is updated according to a namelist parameter (e.g. cloud derived variables, and the frequency of activation of the radiation scheme), (3) instantaneous values which are computed only at the time step when the output is written out (see the figure 1 for more details).

The additional pressure interpolated variables are provided in Table 2. At the same time, in order to do not overload WRF execution, the section of the code with the pressure interpolation has been also modified. Now the interpolation is computed at

15 its specified output time step, and these CORDEX added variables will be gathered in the 23rd auxiliary WRF output file with the standard name `wrfpres_d<domain>_<date>` (in WRF's namelist notation).

The different statistical variables are computed using the values during the internal integration of the model. The values of these statistics are initialized (as shown in figure 1) at the first time step after output time. More details on how certain diagnostic variables have been integrated, and how they have been implemented in WRF are provided in the following sections.

Furthermore, a series of plots accompanying different definitions of the diagnostic variables are presented as well. The intention of these figures is to illustrate the consistency of the implemented diagnostics. These preliminary outcomes are not for validation purposes, but rather to show that the diagnostic variable have been correctly introduced. A complete analysis in order to find the most accurate methodology for the calculation of a certain diagnostic would require devoted climate simulations, and enough observations to validate them. Such validation is out of the scope of this study. We do select 'more appropriate'



**Figure 1.** Calculation of diagnostics according to the time step. Each kind of variable (see Tables in appendix B)

compute variables required at each time step [$a$ and $s$ kind variables]

IF (it == schemefreq)

  compute variable related to the scheme [$t$ kind variables]

ELSE

  keep value of the previous time step [$t$ kind variables]

END IF

IF (MOD(it,outputfreq) == 0)

  initialize statistics values [$s$ kind variables]

    $\chi flux = \chi(it)$

    $\chi min = \chi(it)$

    $\chi max = \chi(it)$

    $\chi mean = \chi(it)$

ELSE

  compute statistics values [$s$ kind variables]

    $\chi flux = \chi flux + \chi(it)$

    IF $(\chi(it) < \chi min) \rightarrow \chi min = \chi(it)$

    IF $(\chi(it) > \chi max) \rightarrow \chi max = \chi(it)$

    $\chi mean = \chi mean + \chi(it)$

END IF

IF (output_time)

  compute variables required only at output time [$o$ kind variables]

  finish certain statistics [$s$ kind variables]

    $\chi flux = \chi flux/(Nsteps \times dt)$

    $\chi mean = \chi mean/Nsteps$

END IF





options based on the experience within the scientific community or according to the CORDEX specifications. These options are set as the 'default' options within the namelist.

One should be aware that certain diagnostics use variables for their calculation which might only be available when specific physical schemes are selected. When this happens, zero values are returned. This undesired outcome is, when possible, fixed
by using a 'generic' definition of the diagnostics.

## 3.2 Core variables

The 'Core' variables are the mandatory ones required by CORDEX. Most of them are standard fields and therefore tend to require simple calculations from the currently available variables from the WRF model. These variables are obtained by setting the pre-compilation flag `CORDEXDIAG`, and will appear in two different files: 3D variables at pressure levels (the WRF model
internally interpolate them since it uses the $\eta$ coordinate in the vertical) will appear in the output file with the 23rd stream (mainly called `wrfpress`), and the 2D variables in the module's output file `wrfcdx`.

### 3.2.1 3D at pressure-levels

These are additional variables which have been added into the WRF pressure-level integration module. Their values will be written in the 23rd output stream as a complement to the one currently available. All of them are instantaneous values.

**hus: humidity**

3D atmospheric specific humidity (hus)[1] and relative humidity (hur) are computed at the standard model $\eta$ levels. Specific humidity is simply obtained from water vapor mixing ratio using equation 1 (named $QVAPOR$ in WRF). Relative humidity can be obtained following the Clausius-Clapeyron formula and its approximation from the well-known *August-Roche-Magnus* formula for saturated water vapor pressure $e_s$,

$$hus = \frac{QVAPOR}{QVAPOR+1} \tag{1}$$

$$e_s = 6.1094 * e^{\frac{17.625*tempC}{tempC+243.04}} \tag{2}$$

$$w_s = \frac{0.622 * es}{presshPa - es} \tag{3}$$

$$hur = \frac{QVAPOR}{ws*1000.} \tag{4}$$

with $tempC$: being temperature in degree Celsius ($°C$), $presshPa$: pressure ($hPa$), $e_s$: saturated water vapor pressure ($hPa$),
$w_s$: saturated mixing ratio ($kgkg^{-1}$).

**press: air-pressure**

WRF model integrates the perturbation of the pressure field from a reference one. Thus to obtain the full pressure at standard model $\eta$ levels, it is required to combine two different fields as shown in equation 5,

---

[1]from the AMS glossary http://glossary.ametsoc.org/wiki/Specific_humidity





$$press = P + PB \tag{5}$$

where $PB$: WRF base pressure ($Pa$), $P$: WRF perturbation pressure ($Pa$).

**ta: air-temperature**

This variable states for the 3D atmospheric temperature on standard model $\eta$ levels. WRF model equations are based on the

perturbation of potential temperature, therefore a conversion to actual temperature is required, and it is performed as indicated

by equation 6,

$$ta = (T + 300)\left(\frac{P + PB}{p0}\right)^{R/C_p} \tag{6}$$

where $T$: WRF 3D temperature output (which is as potential temperature perturbation from the base value, which in WRF

equals to 300 $K$), $p0$: the pressure reference (100000 $Pa$).

**ua/va: Earth roteted wind components**

These variables state for the 3D atmospheric wind components following Earth coordinates on standard model $\eta$ levels. WRF

model equations use the Arakawa-C staggered grid with wind components following the grid direction. In order to get actual

winds following the Earth geographical coordinates, a transformation shown in equation 7 is required,

$$
\begin{cases}
U_{unstg}(1:dimx, 1:dimy) = 0.5[U_{stg}(1:dimx-1, 1:dimy) + U_{stg}(2:dimx, 1:dimy)] \\
V_{unstg}(1:dimx, 1:dimy) = 0.5[V_{stg}(1:dimx, 1:dimy-1) + V_{stg}(1:dimx, 2:dimy)]
\end{cases}
$$

$$
\begin{cases}
ua = U_{unstg}cosa - V_{unstg}sina \\
va = U_{unstg}sina + V_{unstg}cosa
\end{cases}
\tag{7}
$$

where $U_{unstg}$: unstaggered WRF eastward wind ($ms^{-1}$), $V_{unstg}$: unstaggered WRF northward wind ($ms^{-1}$), $U_{stg}$: x-staggered

WRF eastward wind ($ms^{-1}$), $V_{stg}$: y-staggered WRF northward wind ($ms^{-1}$), $cosa$: local cosine of map rotation (1), $sina$,

local sine of map rotation (1).

**zg: geopotential height**

As in the case of air-pressure, WRF model also integrates the perturbation of the geopotential field from a reference or base

one. Thus to obtain the full geopotential height on standard model $\eta$ levels, it is required to combine the two WRF fields as it

is shown in equation 8,

$$zg = PH + PHB \tag{8}$$

where $PHB$: WRF base geopotential height ($m^2 s^{-2}$), $PH$: WRF perturbation of the geopotential height ($m^2 s^{-2}$).



### 3.2.2  2-dimensional

These variables required CORDEX are 2D fields. Some of them are diagnosed as a combination of 3-dimensional variables, some are required as instantaneous values, and others as statistics. The fact that the module provides 2D variables online using 3D fields, shows one more key advantage of the module related to disk space. Thanks to these online calculations when using

5   the module, a user does not need anymore to store large amount of 3D data from the model in order to post-process them. This reduces the requirements for free disk space by a factor of around 2.

**pr, prc, prl, prsh, prsn: precipitation fluxes**

Total precipitation flux (pr) is computed as the sum of all types of precipitation fields in the model accumulated along the 9th stream output frequency (9freq) divided by this accumulated time (9freq), as it is shown in equation 9,

$$pr = \frac{\sum\limits_{it}^{9freq} RAINCV(it) + RAINNCV(it) + RAINSHV(it)}{Nsteps \times \delta t} \tag{9}$$

where $RAINCV$: instantaneous precipitation from cumulus scheme ($kgm^{-2}$), $RAINNCV$: instantaneous precipitation from microphysics scheme ($kgm^{-2}$), $RAINSHV$: instantaneous precipitation from shallow-cumulus scheme ($kgm^{-2}$), $Nsteps$: number of time steps, and $\delta t$: time step length ($s$) to achieve the 9th stream frequency output time ($9freq = Nsteps \times \delta t$).

Each individual precipitation flux is also provided as:

$$\begin{cases} prc = \dfrac{\sum\limits_{it}^{9freq} RAINCV(it)}{Nsteps \times \delta t} \\ prl = \dfrac{\sum\limits_{it}^{9freq} RAINNCV(it)}{Nsteps \times \delta t} \\ prsh = \dfrac{\sum\limits_{it}^{9freq} RAINSHV(it)}{Nsteps \times \delta t} \end{cases} \tag{10}$$

Solid precipitation flux (prsn) only accounts for frozen precipitation. Depending on the selected micro-physics scheme chosen in the 'namelist', this variable might account for the precipitation of snow, graupel and hail. It is computed as it is shown in equation 11,

$$prsn = \frac{\sum\limits_{it}^{9freq} prins(it) \times SR(it)}{Nsteps \times \delta t} \tag{11}$$

20   where $prins$: instantaneous total precipitation (previously obtained), $SR$: fraction of solid precipitation (variable included in WRF).

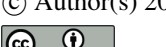



### Radiative flux

Surface upwelling shortwave radiation flux (rsus) and surface upwelling longwave radiation flux (rlus) are understood as the shortwave and longwave radiation from Earth's surface. They are directly provided by radiation schemes CAM[2] and RRTMG[3] (`sw_ra_scheme = 3,4`) as instantaneous variables `swupb` and `slupb`. When there is no use of such schemes, it is recommended to use the 'generic' definition instead (rsusgen, rlusgen, see in next section). Statistical retrieval for the surface fluxes follows the same methodology as for the precipitation fluxes.

### sund: duration of sunshine

This variable accounts for the the sum of the time for which the direct solar irradiance (downwelling short-wave radiation, rsds) exceeds $120\ Wm^{-2}$ (WMO, 2010a) implemented following equation 12. In order to provide an example of the correct implementation of this diagnostics preliminary results are shown in figure 2. The figure shows the 'sund' values and compare them with the incoming solar radiation. It is shown how the 'sund' values vary accordingly to the moment of the day with zero values during night (left panel) or persistent totally cloud covered regions (map at the right panel)

$$sund = \sum_{it}^{9freq} \delta t [SWDOWN(it) \geq 120 Wm^{-2}] \tag{12}$$

where $SWDOWN$: downward shortwave radiation ($Wm^{-2}$), $\delta t$: time-step length ($s$).

### tauuv: surface downward wind stress

Instantaneous surface downward wind stress at 10m accounts for the force that winds exerts on the Earth's surface. It is implemented following the equation 13

$$tauv = \left(C_D uas^2, C_D vas^2\right) \tag{13}$$

where, $C_D$: drag coefficient (1), $uas$: Earth-rotated eastward 10 m surface wind ($ms^{-1}$), $vas$: Earth-rotated northward 10 m surface wind ($ms^{-1}$). The drag coefficient is non-zero only for certain options of the surface layer physics (`sf_sfclay_physics` parameter in the namelist): `1` (MM5-similarity), or `5` (MYNN surface layer). A 'generic' formulation has been introduced when these schemes are not used.

### psl: sea level pressure

This variable accounts for the instantaneous pressure extrapolated to the sea level. It represents the value of the pressure without the presence of orography. In order to provide a framework ready to implement different methodologies, three different methods have already been implemented. The choice of the method can be controlled by a new `namelist.input` parameter labeled `psl_diag` in `cordex` section. The implemented methods are:

---

[2]Community Atmosphere Model

[3]Rapid Radiative Transfer Model



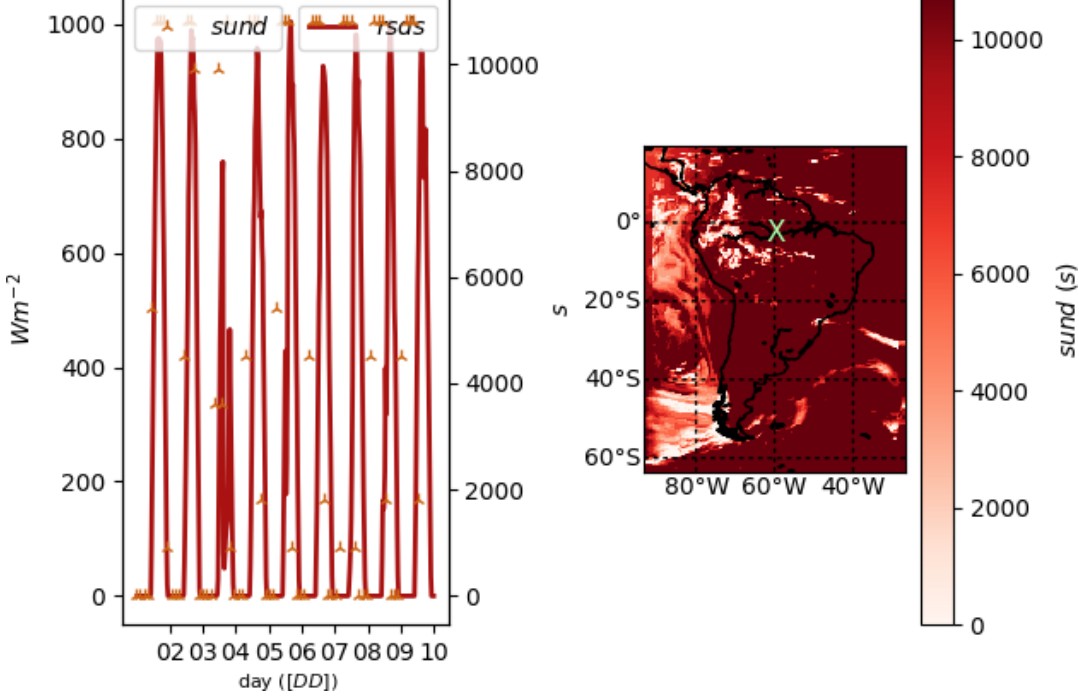

**Figure 2.** Temporal evolution at $S\ 62°\ 4'\ 38.00", 4°\ 58'\ 55.51"W$ (left panel) of shortwave downward radiation ($rsds$, red line, left y-axis) and sunshine duration for a 3-hourly 9freq ($sund$, stars, right y-axis). Sunshine length map on 2012-12-09 at 15 UTC. X denotes position of the temporal evolution

- The hydrostatic-Shuell method (Stackpole and Cooley, 1970) already implemented in the the module `phys/module_diag_afwa.F`, assuming a constant lapse-rate of $-6.5\ Kkm^{-1}$, selected when in WRF 'cordex' namelist section setting the parameter [`psl_diag = 1`]

- The 'ptarget' method (Benjamin and Miller, 1990) that uses smoothed surface pressure and a target upper-level pressure, already implemented in `p_interp.F90` [`psl_diag = 2`]

- The ECMWF method (Yesad, 2015) taken from the Laboratoire de Météorologie Dynamique GCM (LMDZ; Hourdin et al., 2006) scheme from the module `pppmer.F90`, following the methodology by Mats Hamrud and Philippe Courtier from ECMWF [`psl_diag = 3`]

According to the CORDEX specifications, the default method is the ECMWF method. When choosing the 'ptarget' method (`psl_diag = 2`), also the degree of smoothing of the surface using the surrounding nine point average can be chosen by selecting a number of smoothing passes (`psmooth`, default 5), and the upper pressure that has to be used as the target (`ptarget`, default 700 $hPa$).

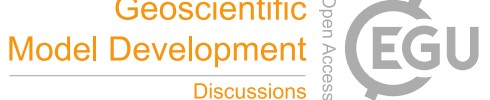



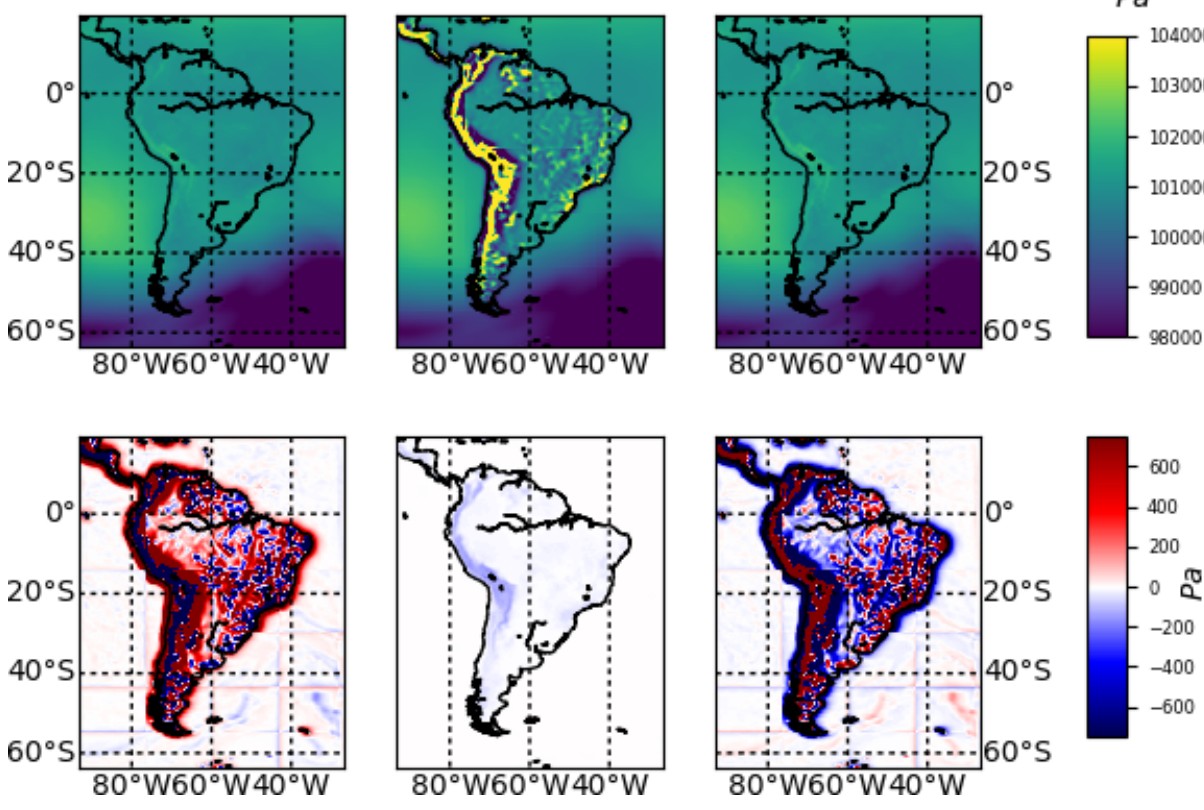

**Figure 3.** Sea level pressure estimates following the hydrostatic-Shuell method at a given time step ($psl^{shuell}$, upper left), ptarget ($psl^{ptarget}$, upper middle) and ECMWF ($psl^{ecmwf}$, upper right). Bottom panels show differences between methods $psl^{shuell} - psl^{ptarget}$ (bottom left), $psl^{shuell} - psl^{ecmwf}$ (bottom middle) and $psl^{ptarget} - psl^{ecmwf}$ (bottom right)

Figure 3 shows the different outcomes applying each method. There are some problems with the p-target method in both psl estimates (mountain ranges can still be inferred) and borders for each parallel process (lines in figures showing differences among methods) when applying the smoothing. Lines showing the limits of the parallel processes appear because one can not obtain the proper values from outside the correspondent tile of the domain associated with the parallel process.

## 5    Cloud derived variables

Four cloud derived variables are required by CORDEX: the total cloudiness (clt) and the cloudiness for each grid point at three different vertical layers above ground (low: $p \geq 680 hPa$, cll; medium: $680 < p \geq 400\ hPa$, clm; high: $p < 400\ HPa$, clh). These cloud diagnostics are provided as mean values.

The module computes these variables taking the cloud fraction of a given grid cell and level as input. The cloud fraction in WRF is computed by the radiative scheme, and is called at a frequency given by `radt` parameter in WRF namelist, which is due to its large computational cost usually larger than the simulations time-step. This determines when cloud fraction is





also actualized to meet the evolved atmospheric conditions. Cloud fractions can be computed in the model using different methodologies. It would be possible to make available these methodologies as another choice in the namelist section and even online compute the cloud fraction at each time-step instead that `radt`. However, in order to be consistent with the radiative cloud effects that the simulation is experiencing, this method was discarded. Thus, the cloud values provided by the module

follow the same frequency of refreshing rate as the one set for radiation in the namelist level.

The most common implementation of 'clt' found in other models (in particular most GCMs) assumes *'random overlapping'*. Random overlapping assumes that adjacent cloud layers are from the same system, hence are maximumly overlapped (Geleyn and Hollingsworth, 1979). In the module, the methodology from the GCM LMDZ was implemented. In this GCM, calculation of the total cloudiness and different layers' cloudiness is done inside the subroutine `newmicro.f90`. The method basically

consists in a product of the consecutive non-zero values of cloud fraction as it is shown in equation 14,

$$zclear = 1, zcloud = 0, ZEPSEC = 1.0 \times 10^{-12} \qquad (14)$$

$$iz = 1 \; to \; dimz \begin{cases} zclear = & zclear\frac{1-MAX[CLDFRAC(iz),zcloud]}{1-MIN[zcloud,1.-ZEPSEC]} \\ clt = & 1 - zclear \\ zcloud = & CLDFRAC(iz) \end{cases}$$

where $CLDFRAC$: cloud fraction (1) at each vertical level, $zclear$: clear-sky value (1), $zclout$: cloud-sky value (1), $ZEPSEC$: value for very tiny number. Same methodology as in equation 14 is applied for the diagnostic of 'clh', 'clm' and 'cll' but split-

ting by corresponding pressure intervals. The figure 4 illustrates the result of the implementation and compare the results with the actual values of the cloud fraction (a and b panels) as well as the different cloud distribution (panels c to f).

**Wind derived variables**

CORDEX requires two wind-derived diagnostics: the daily maximum near-surface wind speed of gust (wsgsmax) and the daily maximum wind speed of gust at 100m above ground (wsgsmax100). These variables can not be retrieved by post-processing

the standard output since they require the combination of different variables (some of them not provided) and have to be provided as a maximum value. The module provides different ways to compute them under certain limitations.

wsgsmax: maximum near-surface wind speed of gust

The wind gust accounts for the wind from upper levels that is projected to the surface due to instability within the planetary boundary layer. In the current version of the module two complementary methods of diagnosing the variable (resultant winds are Earth-rotated) have been implemented. The choice between the two methods can be done by switching a new

`namelist.input` parameter labeled `wsgs_diag` (in `cordex` section), with the default value set to `1`. The implemented methods are:

– The Brasseur method (Brasseur01), [`wsgs_diag = 1`]: An implementation of wind gust considering Turbulent Kinetic Energy ($TKE$) estimates and stability defined by virtual temperature ($\theta_v$) as indicated in equation 15 following Brasseur (2001). Implementation is adapted from a version already introduced in the CLimate WRF (clWRF,





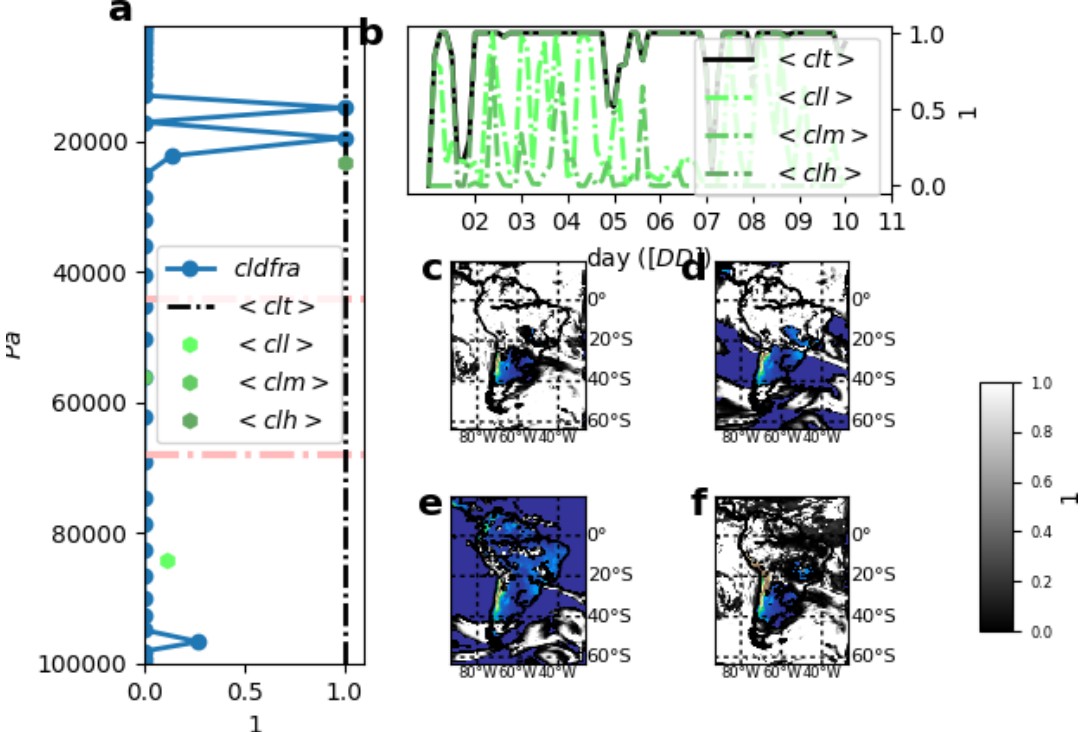

**Figure 4.** Vertical distribution of cloud fraction and the different cloud layers on 2012-12-09 at 15 UTC at $S\,62°\,4'\,38.00"$, $4°\,58'\,55.51"W$ (a): cloud fraction ($cldfra$, full circles with line in blue), mean total cloud fraction ($clt$, vertical dashed line), mean low-level cloud fraction ($cll\ p \geq 680\ hPa$, dark green hexagon), mean mid-level ($clm\ 680 < p \geq 440\ hPa$, green hexagon), mean high-level ($clh\ p < 440\ hPa$, clear green hexagon). Temporal evolution of cloud layers at the given point (b). Map of $clt$ with colored topography beneath to show-up cloud extent (c), map of $clh$ (d), map of $clm$ (e) and map of $cll$ (f)

http://www.meteo.unican.es/wiki/cordexwrf/SoftwareTools/ClWrf; Fita et al., 2010),

$$\frac{1}{z_p}\int_0^{z_p} TKE(z)dz \geq \int_0^{z_p} g\frac{\Delta\theta_v(z)}{\Theta_v(z)}dz \qquad (15)$$

where $TKE$: Turbulent Kinetic Energy ($m^2 s^{-2}$), $\theta_v$: virtual potential temperature ($K$). $z_p$ height of the considered parcel ($m$, satisfying 15), $\Delta\theta_v(z)$: variation of $\theta_v$ over a given layer ($Km^{-1}$).

5    – The AFWA method [`wsgs_diag = 2`]: An implementation adopted from the WRF module `module_diag_afwa.F` which calculates the wind gust that only occurs as a blending of upper-level winds $zagl$ (around 1 km above ground; $zagl(k_{1000}) \geq 1000\,m$, see equation 16) when precipitation intensity per hour is above a given maximum value, $prate_{hr}^{mm} \geq$

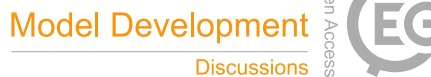

$50\ mmh^{-1}$

$$
\begin{aligned}
\boldsymbol{va}_{1km} &= \boldsymbol{va}(k_{1000}-1)+[1000-zagl(k_{1000}-1)]\frac{\boldsymbol{va}(k_{1000})-\boldsymbol{va}(k_{1000}-1)}{zagl(k_{1000})} \\
\gamma &= \frac{150-prate_{hr}^{mm}}{100} \\
\boldsymbol{va}_{blend} &= \boldsymbol{vas}\gamma+\boldsymbol{va}_{1km}\times(1-\gamma)
\end{aligned}
\tag{16}
$$

5      where $va$ : air wind $(ms^{-1})$, $zagl$: height above ground $(m)$, $k_{1000}$: vertical level at which $zagl$ is equal or above 1000 m. $prate_{hr}^{mm}$: hourly precipitation rate $(mmh^{-1})$, $va_{blend}$: blended wind $(ms^{-1})$

The two methods provide wind gust estimation (WGE) from two different perspectives: mechanic and convective. In order to take into account both perspectives, additional variables: $totwsgsmax$ (total maximum wind-gust speed at surface), $totugsmax$ (total maximum wind-gust eastward speed at surface), $totvgsmax$ (total maximum wind-gust northward speed at 10   surface), and $totwsgspercen$ (percentage of time steps along 9freq in which grid point got wind gust (%). Figure 5 shows the outcomes when applying each method. It is shown how wind gust is mainly originated by instability, with a minor impact of heavy precipitation events at the given time. Furthermore, in the bottom panel it is shown how wind-gusts are highly frequent above the sea in comparison to the low frequency above continental flat areas (Andes mountain range exhibits high occurrence of wind gust).

15    wsgsmax100: daily maximum wind speed of gust at 100 m

The calculation of wind gust at 100 m should follow a similar implementation used for calculating the `wsgsmax`, but at 100 m. An extrapolation of such turbulent phenomena would require a complete new set of equations which have not been placed yet. Instead, as a way to overcome it, the estimation of maximum wind speed at 100 m is provided. Provided wind components are also Earth-rotated. Three different methods have been implemented. Two following per-assumed vertical wind 20   profiles (after PhD thesis Jourdier, 2015) and a third one following Monin-Obukhov theory to estimate the wind components above ground. These three methods can be chosen by defining a new namelist parameter labeled `wsz100_diag` (in `cordex` section). Its default value is `1`. The implemented methods are:

– [`wsz100_diag = 1`], following the power-law wind vertical distribution as depicted in equation 17 using the upper-level atmospheric wind speed below ($k_{100}^{<}$) and above ($k_{100}^{>}$) the height above ground of $100\ m$ ($zagl$),

$$
\begin{aligned}
\boldsymbol{va}_{100} &= \boldsymbol{va}(k_{100}^{>})\left(\frac{100.}{zagl(k_{100}^{>})}\right)^{\alpha_{x,y}} \\
\alpha_{x,y} &= \frac{\ln(\boldsymbol{va}(k_{100}^{>}))-\ln(\boldsymbol{va}(k_{100}^{<}))}{\ln(zagl(k_{100}^{>}))-\ln(zagl(k_{100}^{<}))}
\end{aligned}
\tag{17}
$$





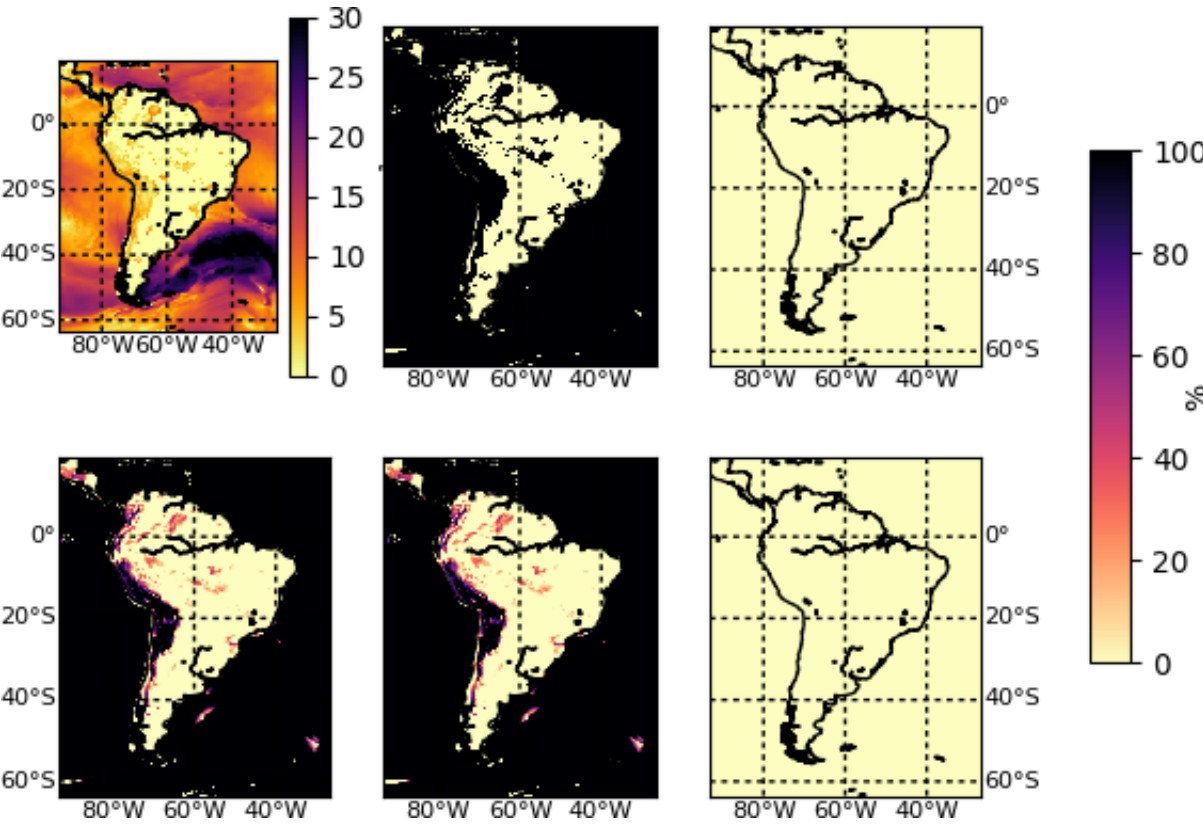

**Figure 5.** Near surface wind gust estimates on 2012 December the 9th at 15 UTC. 3h-maximum total wind gust strength ($wsgsmax^{tot}$, top left), percentage of $wsgsmax^{tot}$ following the Brasseur method ($wsgsmax^{b01}$, top middle), percentage following the AFWA-heavy precipitation implementation ($wsgsmax^{hp}$, top right), percentage of time steps where grid point got total wind gust (bottom left), percentage of time steps where grid point got $wsgsmax^{b01}$ (bottom middle), percentage due to $wsgsmax^{hp}$ (bottom right)

– [wsz100_diag = 2], following logarithmic-law wind vertical distribution, as it is depicted in equation 18, using upper-level atmospheric wind speed below ($k_{100}^{<}$) and above ($k_{100}^{>}$) the height above ground of $100\ m$ ($zagl$)

$$\ln(z_0) = \frac{\boldsymbol{va}(k_{100}^{>})\ln(zagl(k_{100}^{<})) - \boldsymbol{va}(k_{100}^{<})\ln(zagl(k_{100}^{>}))}{\boldsymbol{va}(k_{100}^{>}) - \boldsymbol{va}(k_{100}^{<})} \qquad (18)$$

$$\boldsymbol{va}_{100} = \boldsymbol{va}(k_{100}^{>})\frac{\ln(100.) - \ln(z_0)}{\ln(zagl(k_{100}^{>})) - \ln(z_0)}$$

5    – [wsz100_diag = 3], following Monin-Obukhov theory. User should keep in mind that this method is not useful for heights larger than few decimeters ($z > 80\ m$). The wind at given height is extrapolated following turbulent mechanisms. As it is shown in equation 19, surface wind speed is used as surrogate to estimate 100 m wind direction ($\theta_{10} = \tan^{-1}(uas, vas)$, without considering Eckman pumping, or other effects on wind direction). In this implementation $u_*$ in similarity theory is taken as WRF estimates UST, Monin-Obukhov length ($\mathcal{L}_O$) as the WRF values RMOL,




roughness length ($z_0$) as WRF thermal time-varying roughness length ZNT,

$$
\begin{aligned}
wss_{100} &= \frac{UST}{\kappa}\left(\ln\left(\frac{100}{z_0}\right) + \Psi_M\left(\frac{100}{\mathcal{L}_O}\right)\right) \\
\mathcal{L}_O &= \frac{-UST^3 T_v}{\kappa g Q_0} \ (Obukhov\ length) \\
\Psi_M\left(\frac{z}{\mathcal{L}_O}\right) &\begin{cases} \frac{4.7z}{\mathcal{L}_O} & \frac{z}{\mathcal{L}_O} > 0 \quad (sTable) \\ \ln\left[\left(\frac{1+X^2}{2}\right)\left(\frac{1+X}{2}\right)^2\right] - 2\tan^{-1}(X) + \frac{\pi}{2} & \frac{z}{\mathcal{L}_O} < 0 \quad (unsTable) \end{cases} \\
X &= \left(1 - \frac{15z}{\mathcal{L}_O}\right)^{1/4}
\end{aligned}
\tag{19}
$$

$$
\theta_{10} = atan\left(\frac{V10}{U10}\right) \boldsymbol{va}_{100} = \begin{cases} ua_{100} = wss_{100}\cos(\theta_{10}) \\ va_{100} = wss_{100}\sin(\theta_{10}) \end{cases}
$$

$wss_{100}$: wind speed at 100 m ($ms^{-1}$), where $\Psi_M$: stability function after (Businger et al., 1971), $UST$: $u^*$ in similarity theory ($ms^{-1}$), $z_0$: roughness length ($m$), $U10, V10$: 10-m wind speed, $theta_{10}$: 10-m wind speed direction ($rad$), $ua_{100}$: 100 m eastward wind speed, $va_{100}$: 100 m northward wind speed (note the absence of correction in wind direction to Ekman pumping or other turbulence effects)

The user can select another height for the estimation by providing the new parameter `z100m_wind` with the value other than $100\ m$ (default value).

Figure 6 shows different preliminary results using the three different approximations. It is illustrated (a panel) how wind-gusts are larger than the 10-m diagnostic winds, and also the difference is larger when using Monin-Obukhow method compared to the two others methods. Certain problems (too small Monin-Obukhov length) are recognized when applying Monin-Obukhov for extrapolating wind at 100 m, which is shown in panel b, where wind gusts appear to be strong as $80\ ms^{-1}$. Therefore a user is advised to use this method with care.

**Vertically integrated variables**

The instantaneous vertically integrated amount of water vapor (prw), liquid condensed water species (clwvi), ice species (clivi), graupel (clgvi), hail (clhvi) are the vertically integrated amounts of each specie along the vertical column (density weighted) over each grid point. They are provided using the same implementation as those in `p_interp.F` - WRF tool for vertical interpolation. The general equation following WRF standard variables is:

$$
clvivar = \frac{MU + MUB}{g} \sum_{iz=1}^{e\_vert} WRFVAR[iz](DNW[iz])
\tag{20}
$$

where $clvivar$: the column vertically integrated variable's CF-compliant name (prw, clwvi, clivi, clgvi, or clhvi), $MU$: pertur-bation dry air mass in column ($Pa$), $MUB$: base-state dry air mass in column ($Pa$), $g$: gravity ($ms^{-2}$), $e\_vert$: total number of vertical levels, $WRFVAR$: the water species' mixing ratio at each sigma level ($kgkg^{-1}$), $DNW$: difference between two consecutive full-eta levels ($-$). Table 3 indicates the $WRFVAR$ names associated with the $clvivar$ names.





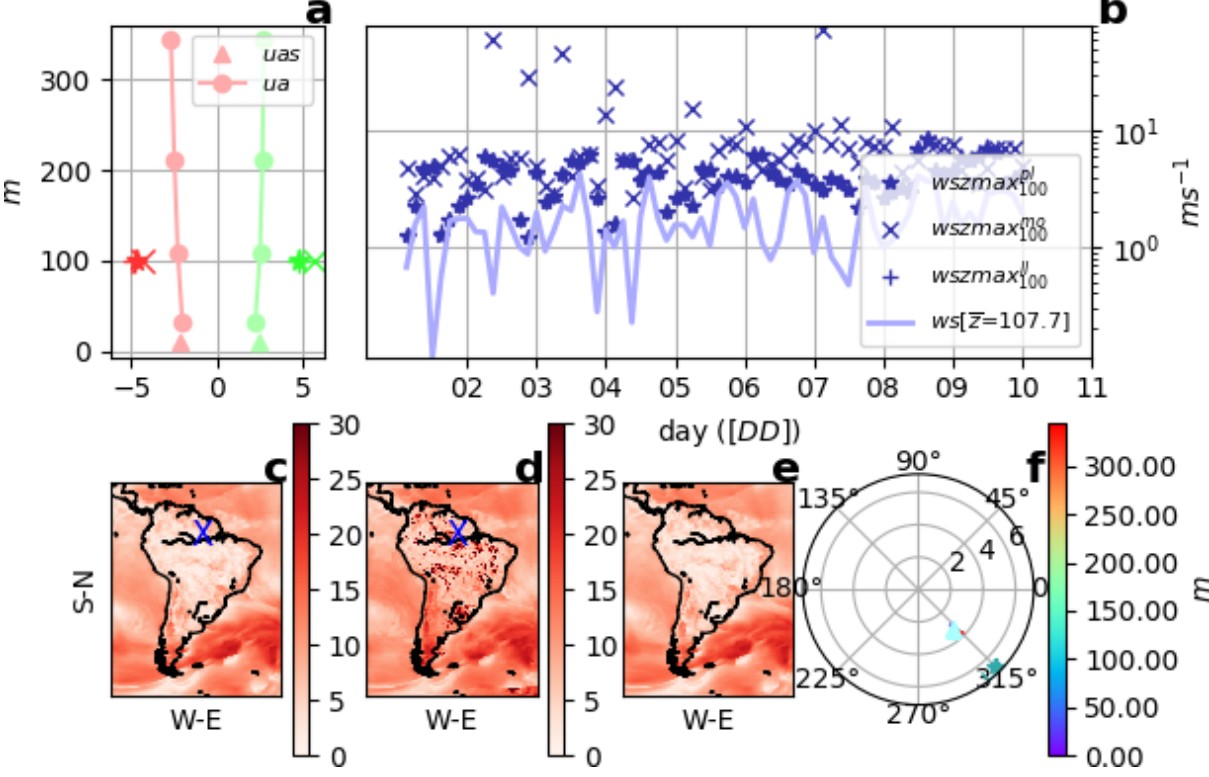

**Figure 6.** 100 m wind estimates. Comparison between upper-level winds and estimation on 2012-12-09 at 15 UTC at $S\ 62°\ 4'\ 38.00"$, $4°\ 58'\ 55.51"W$ (a): 3h-maximum eastward wind (red) at 100 m by power-law ($uzmax^{pl}$, star marker), Monin-Obukhov theory ($uzmax^{mo}$, cross) by logarithmic law ($uzmax^{ll}$, sum) 10-m wind value ($uas$, filled triangle) and upper-level winds ($ua$, filled circles with line), also for the northward component (green). Temporal evolution of wind speed (b) with all approximations and upper-level winds at the closest vertical level at 100 m (on log-y scale). Maps of three estimations: power-law (c), Monin-Obukhov (d), logarithmic-law (e) with the blue cross showing the point of previous figures. Vertical distribution of winds at the given point in Wind rose-like representation (f)

**Table 3.** Mixing ratio associated with column integrated variables

| name | WRF species | description |
|---|---|---|
| prw | QVAPOR | water vapor mixing ratio |
| clwvi | QCLOUD+QRAIN | condensed water and rain mixing ratio |
| clivi | QICE+QSNOW+QGRAUPEL+QHAIL | ice, snow, graupel and hail mixing ratio |
| clgvi | QGRAUPEL | graupel mixing ratio |
| clhvi | QHAIL | hail mixing ratio |





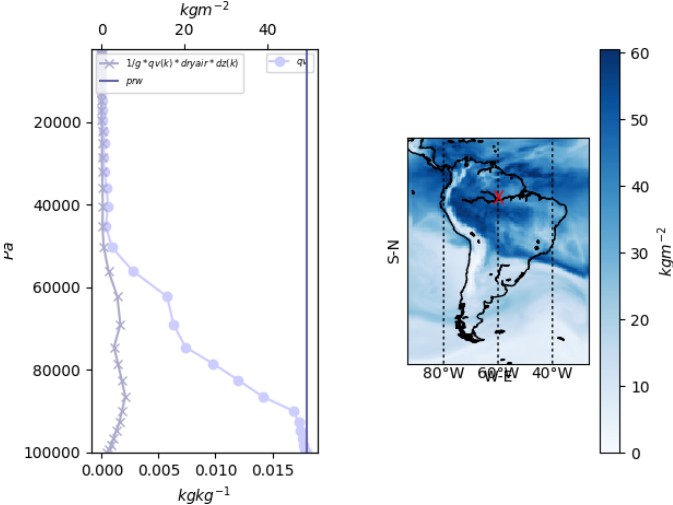

**Figure 7.** On 2012-12-09 at 15 UTC at $S\ 62°\ 4'\ 38.00"$, $4°\ 58'\ 55.51"W$ (left): water path ($prw$, vertical straight line in $mm$ top x-axis), vertical profile of water vapour ($qv$, line with full circles in $kgkg^{-1}$ bottom x-axis), water path at each level (line with crosses). Map of water path (right) on 2012-12-09 at 15 UTC, red cross shows where the vertical accumulation is retrieved

Note that clgvi and clhvi are part of the 'Tier1' level and are only accessible if pre-compilation variable `CDXWRF` is set to 1. See section 2.1 for more detail. In order to provide an example of the correct computation of the diagnostics, results at a given grid point are shown in figure 7. It is shown that the total precipitable water (prw) correctly corresponds to the density weighted vertical integration of the water content along the column of air.

5 **evspsblpot: potential evapotranspiration**

This variable represents the evaporative demand of the atmosphere. It is computed following the common method included in the GCMs. One of the first proposed methods was provided by Manabe (1969). Different iterations later, some corrections have been proposed to the initial methodology in order to overcome its deficiencies (e.g. see Barella-Ortiz et al. (2013) for 10 an intercomparison among different methods). It is provided as a statistical flux. In the module, the original an a correction methodology are provided and selected via the namelist parameter `potevap_diag`, with 2 being the default value:

  – bulk method [`potevap_diag = 1`]: this method corresponds to the initial one proposed in Manabe (1969). It basically consists in a difference between a supposed saturated air at the surface temperature and the humidity of the atmosphere as it is depicted in equation 21,

15 $$qc = C_D\sqrt{U10^2 + V10^2}$$

$$evspsblpot_{bulk} = \rho(1)qc\left[ws(ts) - QVAPOR(1)\right] \tag{21}$$





where $ws(ts)$: saturated air at ts ($kgkg^{-1}$), $qc$: surface drag coefficient ($ms^{-1}$), $TSK$: surface temperature ($K$), $ws(ts)$: saturated air by surface temperature ($kgkg-1$) based on August-Roche-Magnus approximation, $press$: air pressure ($Pa$), $U10, V10$: 10 m wind components ($ms^{-1}$), $QVAPOR$: 3D water vapour mxing ratio ($kgkg^{-1}$), $C_D$: drag coefficient ($-$, only available from MM5-similarity and MYNN surface layer schemes, otherwise is zero).

5     – Milly92 method [potevap_diag = 2]: this method makes a correction of the bulk diagnostic by introducing a Milly's correction parameter $\xi$, which accounts for other atmospheric-related phenomena (Milly, 1992). It is explained in equation 22, and its implementation is similar to the one present in ORCHIDEE model (Organising Carbon and Hydrology In Dynamic Ecosystems, http://orchidee.ipsl.fr/, de Rosnay et al. (2002). The implementation is retrieved from the module `src_sechiba/enerbil.f90`,

$$\beta = \frac{sfcevap}{evspsblpot_{bulk}}$$
$$\partial_T ws(T) = \frac{ws[T(1) + 0.5] - ws[T(1) - 0.5]}{2 \times 0.5}$$
$$\xi = \frac{L\rho(1)qc\partial_T ws(T)(1-\beta)}{4 EMISS CtBoltzman T(1)^3 + \rho(1)Cpqc + L\rho(1)qc\partial_T ws(T)\beta}$$
$$evspsblpot_{Milly92} = evspsblpot_{bulk} \frac{1}{1+\xi} \tag{22}$$

where $\beta$: Moisture availability function, $sfcevap = QFX$ surface evaporation ($kgm^{-2}s^{-1}$) from $QFX$: surface mois-
ture flux ($kgm^{-2}s^{-1}$), $L$: latent heat of vaporization, $EMISS$: emissivity (1), $CtBoltzman$: constant of Stefan-Boltzman, $Cp$: specific heat of air, $\partial_T ws(T)$: derivative of saturated air at temperature of the first atmospheric layer ($kgkg^{-1}$) usin numerical 1st order approximation

See in figure 8 an example of the differences between both implementations. It shows how important the correction introduced by Milly is and the effect on the diurnal cycle. Basically, the correction permits potential evapotranspiration during
night time and reduces its strength at noon (18 UTC corresponds to 12 local time). There is a generic diagnostic to overcome boundary layer scheme dependency in the calculation of the drag coefficient (see below in generic variables).

### 3.3 Generic variables

Some of the diagnostics required by CORDEX depend on the approximations, equations, methodologies and observations used to compute them. This makes model intercomparison very difficult, since values might differ from one implementation to
25 another. To overcome in a way this issue, a series of variables are also provided in a 'generic' form (when possible) meaning that they are obtained directly from standard variables. Thus, these generic forms of the diagnostics become 'independent' on the model's implementation.

**cdgen: generic surface drag coefficient**

Computation of the instantaneous drag coefficient at the surface depends on the selected surface scheme. In order to avoid this
scheme dependency, a generic calculation of the coefficient has been introduced as in equation 23 following Garratt (1992),





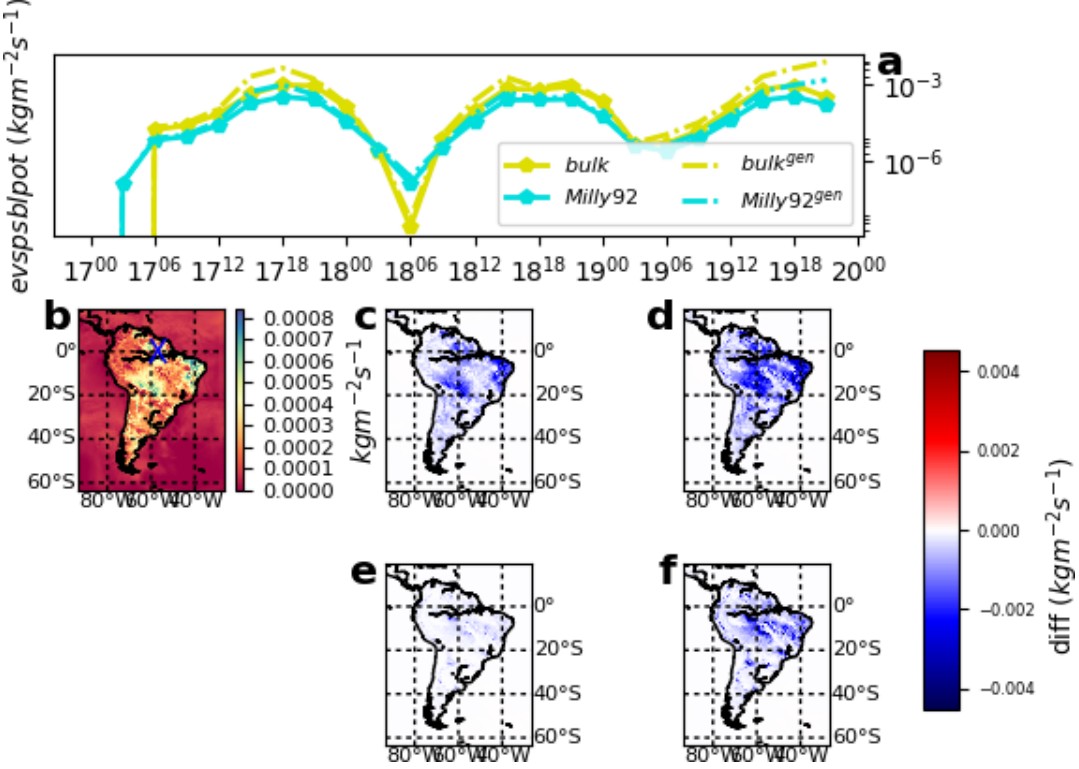

**Figure 8.** Evolution (a, in y-log scale) of potential evapotranspiration by bulk (yellow) and Milly92 (blue) generic (dashed lines) methods at S $4°\ 58'\ 55.524"$, $62°\ 4'\ 37.92"\ W$ (blue cross in b). On 2015-11-18 15 UTC, potential evapotranspiration following Milly correction (b), differences between both methods (c, $evspblpot_{Milly92} - evspblpot_{bulk}$), differences between both generic methods (d, $evspblpot^{gen}_{Milly92} - evspblpot^{gen}_{bulk}$), differences between Milly method and its generic counterpart (e, $evspblpot_{Milly92} - evspblpot^{gen}_{Milly92}$) and differences between bulk method and its generic counterpart (f, $evspblpot_{bulk} - evspblpot^{gen}_{bulk}$)

$$C_D^{gen} = \left(\frac{UST}{wss}\right)^2 \tag{23}$$

with $UST$: $u^*$ being friction velocity from the similarity theory $(ms^{-1})$, and $wss = \sqrt{U10^2 + V10^2}$: 10-m wind speed $(ms^{-1})$

**tauuvgen: generic surface downward wind stress**

5  Generic surface downward wind stress at 10m is calculated following the equation 24 which uses the generic diagnostics of the drag coefficient.

$$tauvgen = \left(C_D^{gen} uas^2, C_D^{gen} vas^2\right) \tag{24}$$





where $C_D^{gen}$: generic drag coefficient ($-$, see equation 23), $uas$: Earth-rotated eastward 10 m wind component, $vas$: Earth-rotated northward 10 m wind component.

**rsusgen: surface upwelling shortwave radiation**

Surface upwelling shortwave radiation is the shortwave radiation directed from the surface. It is calculated in a generic way as
the reflected shortwave radiation depending on the surface albedo as it is shown in equation 25,

$$rsus = -ALBEDO * SWDOWN \tag{25}$$

Being, $ALBEDO$: albedo (1), $SWDOWN$: downward at surface shortwave radiation ($Wm^{-2}$)

**rlusgen: surface upwelling longwawe radiation**

Surface upwelling longwave radiation is the longwave radiation coming from the surface. It is calculated in a generic way as
the longwave radiation from a black body due to surface temperature following the Stefan–Boltzmann law as it is given in equation 26,

$$rlus = CtBoltzman * EMISS * TSK^4 \tag{26}$$

Being, $CtBoltzman$: the Stefan–Boltzmann constant ($= 5.67051E^{-8}\ Wm^{-2}K^{-4}$), $EMISS$: emissivity (1), $TSK$: skin temperature ($K$)

**evspsblpotgen: generic potential evapotranspiration**

This variable corresponds to the generic definition of potential evapotranspiration ('evspsblpot'). The same two methodologies as in the the regular diagnostic have been implemented. The only difference is that in this case, the generic estimation of the drag coefficient 'cdgen' is used (see equation 23) instead of the one given by the model.

## 3.4   Tier1 variables

These diagnostics are required by CORDEX, but they are not mandatory. They have been also included as a way to fulfill all the CORDEX requirements. These variables require the setting of the pre-compilation flag `CDXWRF` to 1 and performing some complementary modifications in the module's Registry file `registry.cordex`. See section 2.1 for more details.

**zmlagen: generic boundary layer height**

Instantaneous Planetary Boundary Layer (PBL) height is a clear example of model dependence and even scheme dependence
of how a diagnostic is computed. Each PBL scheme has its own assumptions and 'zmla' is computed in a scheme-dependent specific way.

In order to overcome the model/scheme dependence, we implemented a generic formulation for calculating the PBL height as it was done in (García-Díez et al., 2013) after (Nielsen-Gammon et al., 2008). The method consists in defining the height of the PBL as the first level in the mixed layer (ML) where potential temperature exceeds the minimum potential ML temperature
by more than $1.5\ K$. It has been implemented using the definitions given below:





1. Mixed layer depth (MLD) is defined as the model level ($k_{MLD}$) starting from the second model level at which the variation of the mixing ratio ($QVAPOR(k)$, normalized with its value at the first level) exceeds some predefined threshold value ($QVAPOR(1)$): $\frac{|QVAPOR(k_{MLD})-QVAPOR(1)|}{QVAPOR(1)} > \delta QVAPOR$ (here applied a $\delta QVAPOR = 0.1$)

2. Within the MLD the value with the minimum potential temperature is taken as: $\theta min_{MLD} = min[\theta(1),...,\theta(k_{MLD})]$

3. The level of the PBL height ($k_{zmla}$) is the level at which the maximum variation of potential temperature within the MLD exceeds some predefined threshold value: $\theta(k_{zmla}) - \theta min_{MLD} > \delta\theta$, (here $\delta\theta = 1.5\ K$)

4. The PBL height ($zmla$) is obtained using the geopotential height $zg$ at the calculated $k_{zmla}$ level above the ground ($zagl$): $zmla = zagl(k_{zmla}) = zg(k_{zmla})/g - HGT$, with $HGT$ being surface elevation height above sea level.

No general rule has been applied to determine the correct value of $\delta qv$ used to determine MLD. It can be determined by the namelist parameters `zmlagen_dqv` for $\delta qv$ (default value 0.1) and `zmlagen_dtheta` for $\delta\theta$ (default value 1.5 $K$). Comparison of this implementation with the $zmla$ directly provided by WRF's Mellor-Yamada Nakanishi and Niino Level 2.5 PBL scheme (MYNN2.5 Nakanishi and Niino, 2006) is shown in figure 9. In general the generic estimation produces a higher PBL (a panel) with lower values during night (b panel). Spatial distributions between both diagnostics are pretty similar.

**Convective diagnostics**

Diagnostics related to convective activity are: Convective Available Potential Energy (CAPE) which accounts for all the energy that might be released convectively, Convective Inhibition (CIN) which accounts for processes which inhibit the convection, Height of the Level of Free Convection (ZLFC), Pressure at the Level of Free Convection (PLFC), and Lifted Index (LI) which accounts for the temperature difference between the environmental temperature at some higher level in the troposphere and the temperature that a parcel would have if adiabatically lifted at that level. CORDEX requires these values as statistics between output times (9freq in this case)

Since the version V3.6 of WRF, these variables can already be calculated with the module `module_diag_afwa.F` via the `Buoyancy` function. In this version of the module, this is the only available implementation. These vertically integrated diagnostics have a high computational cost. In order to minimize it, they are only computed at output time step by default. However, if a user requires them as statistics (such as capemin, capemax, capemean), then these diagnostics are computed at all time steps. This behavior of the module is regulated via the namelist parameter `convxtrm_diag` (default value is 0, meaning no computation), and by setting the pre-compilation flag `CDXWRF` to 1 and performing some complementary modifications in module's Registry file `registry.cordex`. See section 2.1 for more detail.

## 4 Additional variables

Some variables not required by CORDEX but which may be interesting and useful to the community for wide variety of the purposes have also been added. These variables will be obtained if the pre-compilation flag `CDXWRF` is set to 2 and some additional modifications are made in module's registry file `registry.cordex`. See section 2.1 for more details.



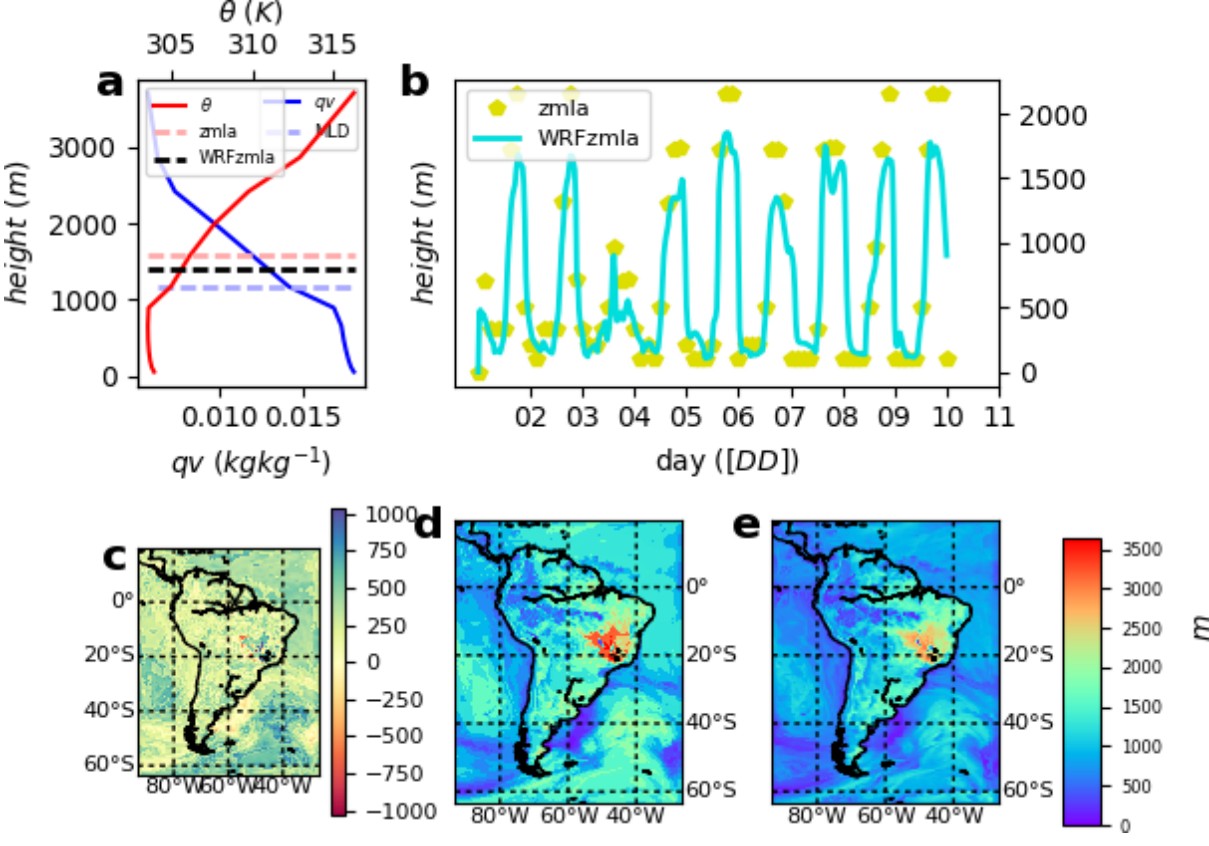

**Figure 9.** Vertical characteristics of the atmosphere on 2012-12-09 at 15 UTC at $S\,62°\,4'\,38.00", 4°\,58'\,55.51"W$ (a): potential temperature vertical profile ($\theta$ K, red line), vertical profile of mixing ratio ($qv\ kgkg^{-1}$, blue line), mixed layer depth ($MLD$, dashed horizontal line at $323.522\ m$), derived boundary layer height ($zmla$, horizontal dashed line at $1007.122\ m$) and WRF derived pbl scheme value ($WRF_{zmla}$ at $903.017\ m$). Comparison of temporal evolutions (b) between derived $zmla$ (yellow stars) and WRF's pbl scheme (blue line). Map of differences between derived and WRF simulated ($zmla - zmla_{WRF}$, c), $zmla$ map (d) and $zmla_{WRF}$ (e)





**tds: dew point temperature**

The dew point temperature (cooler temperature at which air would saturate due to its current moisture content) is calculated following the August-Roche-Magnus approximation as it is shown in equation 27,

$$
\begin{aligned}
\gamma &= \log(hurs) + \frac{b(T2 - 273.15)}{(T2 - 273.15) + c} \\
tds &= \frac{c\gamma}{b - \gamma} + 273.15
\end{aligned}
\tag{27}
$$

where $T2$: 2m temperature ($K$), $hurs$: 2m relative humidity (%, previously computed), $b = 17.625$, $c = 243.04$. This variable is provided as statistics: minimum, maximum and mean in the output.

**Atmospheric water budget**

The water budget accounts for all the dynamics of the water in the atmosphere. This budget is divided in different terms (dynamical and source/sink) accounting for the total mass of water. It can be computed independently for each water specie. The equation for any given water specie is given in equation 28:

$$
\begin{aligned}
TEN_q &= HOR_q + VER_q + MP_q \\
\frac{\partial q_q}{\partial t} &= -V_h \boldsymbol{\nabla} q_q - w \frac{\partial q_q}{\partial z} + SO_q - SI_q
\end{aligned}
\tag{28}
$$

Where $q$ stands for one of the six water species (vapor, snow, ice, rain, liquid, graupel, or hail) concentrations ($kgkg^{-1}$), $V_h$ stands for horizontal wind speed ($ms^{-1}$), $w$ stands for the vertical wind speed ($ms^{-1}$) and $MP$ for the loss or gain of water due to cloud microphysical processes. The term in the left-hand side of the equation represents the water species tendency ($TEN$ or 'PW'), referring to the difference between $q$ at the model's previous time step and at the actual time step, divided by the time step. $TEN$ equals to the horizontal advection ($HOR$ or 'F', first term in right-hand side of the equation), the vertical advection ($VER$ or 'Z', second term in right-hand side) and the sources ($SO$) or sink ($SI$) of atmospheric water due to $MP$. All terms are expressed in $kgkg^{-1}s^{-1}$. However, $SO$ and $SI$ can not be provided because they are micro-physics dependent and make difficult to provide a generic formula for them.

In order to obtain the total column mass of water due to each term (in $mm$), an integration following eq. 29 is applied to each term of eq. 28 (similarly as in 20):

$$
-\frac{1}{g} \int_{p_{sfc}}^{p_{top}} <term> dp
\tag{29}
$$

Following the methodology of Huang et al. (2014) and Yang et al. (2011); Fita and Flaounas (2018) implemented a new module in WRF in order to allow the computation of the water budget terms during model integration. This implementation is provided with the CORDEX module, but variables are only provided as temporal accumulations (within 9freq) and vertical integrations in two forms: total column values and divided by the same layers as the cloud diagnostics (low, medium, high). The accumulation of diabatic heating from the microphysics scheme is provided as a proxy of the sink/sources due to microphysics effects.





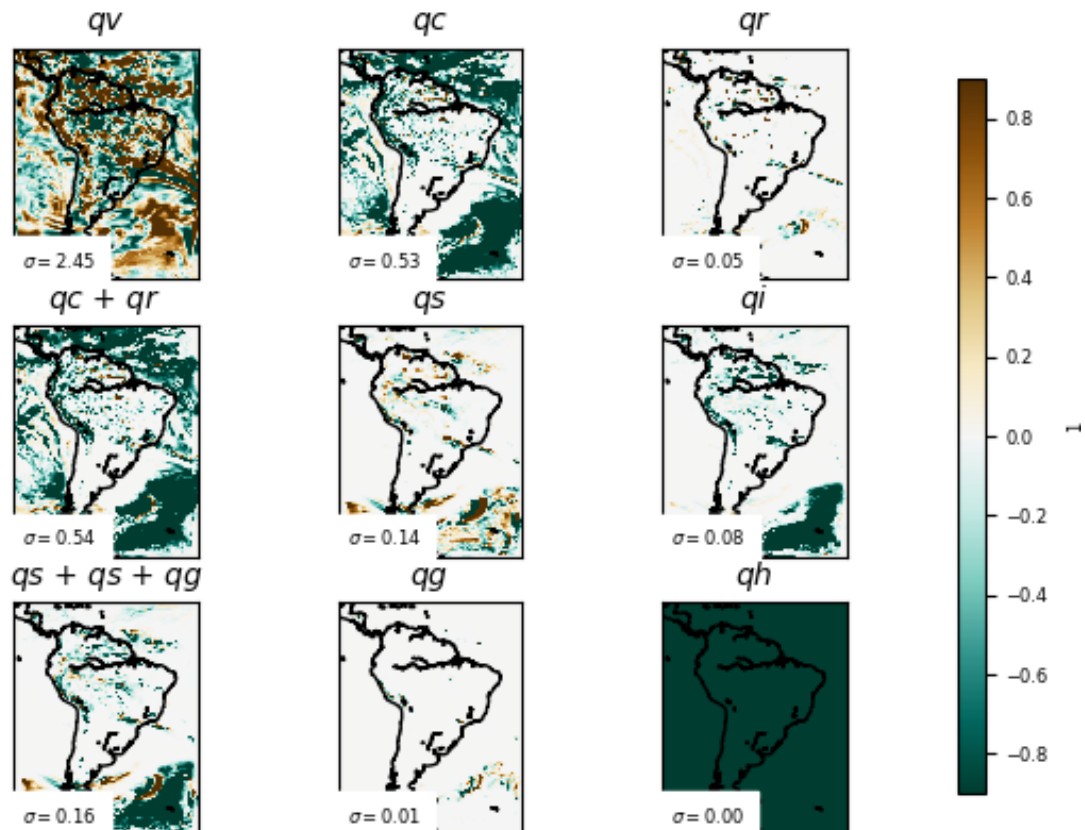

**Figure 10.** Normalized (with the spatial standard deviation of the mapped values, $\sigma$) water budget 3h-accumulated vertically integrated total tendency 'PW' at a given time, for water vapour ($qv$, top left), cloud ($qc$, top middle), rain ($qr$, top right), water condensed species ($qc + qr$, middle left), snow ($qs$, middle middle), ice ($qi$, middle right), water solid species ($qs + qi + qg$, bottom left), graupel ($qg$, bottom middle), hail ($qh$, bottom right). Number on low left corner of the figure correspond to the standard deviation ($\sigma$ in $mm$) value used for the normalization

Preliminary results for all water species are shown in figures 10 and 11. Water vapour exhibits the largest values in both total tendency and horizontal advection. Dynamics of the other water species seems to be highly correlated with the presence of a storm system (lower right corner in the maps) or due to orthographic influences (existence of Andes range can be inferred).

Figures from 12 to 15 show temporal evolution and accumulated maps at a given time for all the water budget terms, decomposed for vapour ($qv$) and snow ($qs$). Accumulated maps are are grouped into vertical levels as it is done with the clouds: $p \geq 68000 \ Pa$, $40000 \leq p < 68000 \ Pa$, $p < 40000 \ Pa$. Largest amounts of the budget terms are mainly found in low (high) levels for water vapour (snow), temporal evolution at a given point show complexity of the water dynamics with the terms compensating each other. It is also shown how contribution to the total diabatic term is large at low levels over the ocean (showing the role of evaporation) and larger at high levels above the continent.

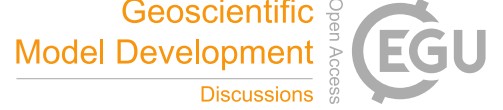

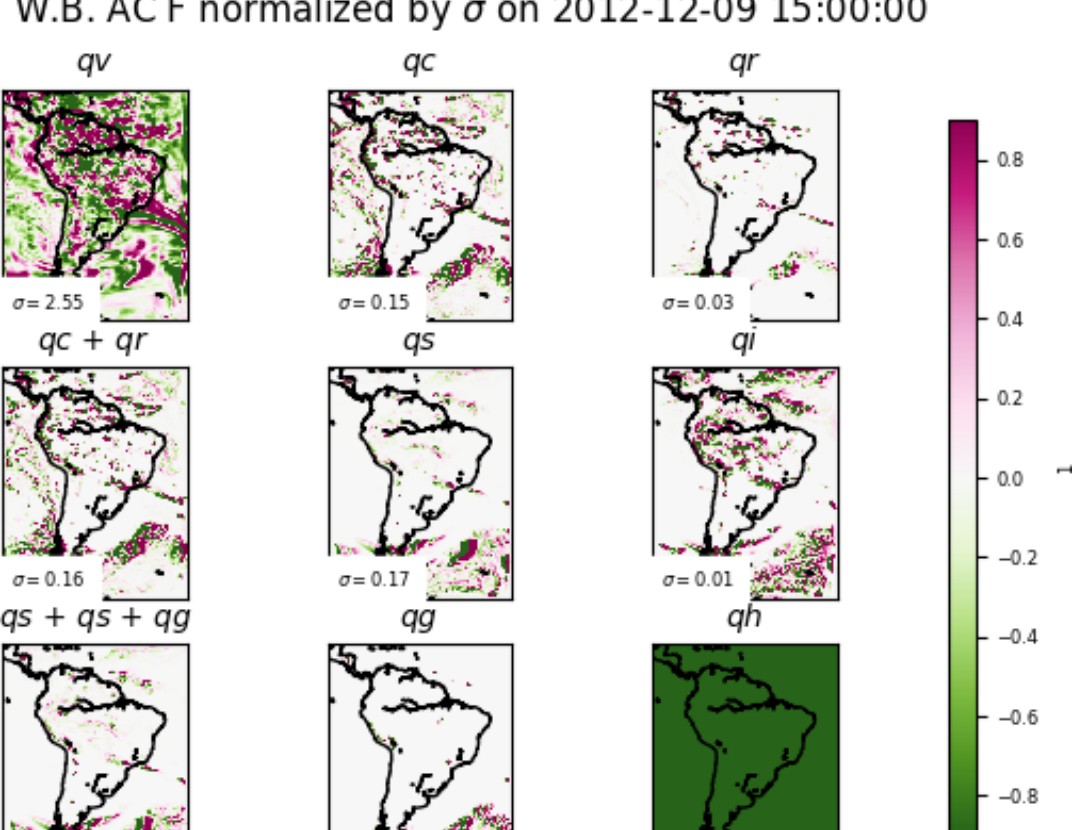

**Figure 11.** As in 10, but for Water budget 3h-accumulated vertically integrated horizontal advection 'F' at a given time

### fogvis: visibility inside fog

Fog is one of major causes of transportation disruption. The horizontal resolutions of state-of-the-art CORDEX activities like FPS_Alps (3 km) open the possibility to explore phenomena such as fog which were impossible to be analyzed in previous experiments. In order to be able to contribute in the analysis of fog phenomena, three different methods to calculate visibility have been introduced. Visibility is used to determine the presence of fog at a given moment. In order to provide a quantity with the density of the fog, only the visibility during a fog event is kept. The three methods are:

- **K84** [`visibility_diag = 1`]: Visibility is computed using liquid water (`QCLOUD`) and ice (`QICE`) concentrations. Following (Bergot et al., 2007), fog appears when there are liquid and/or ice water species at the lowest model





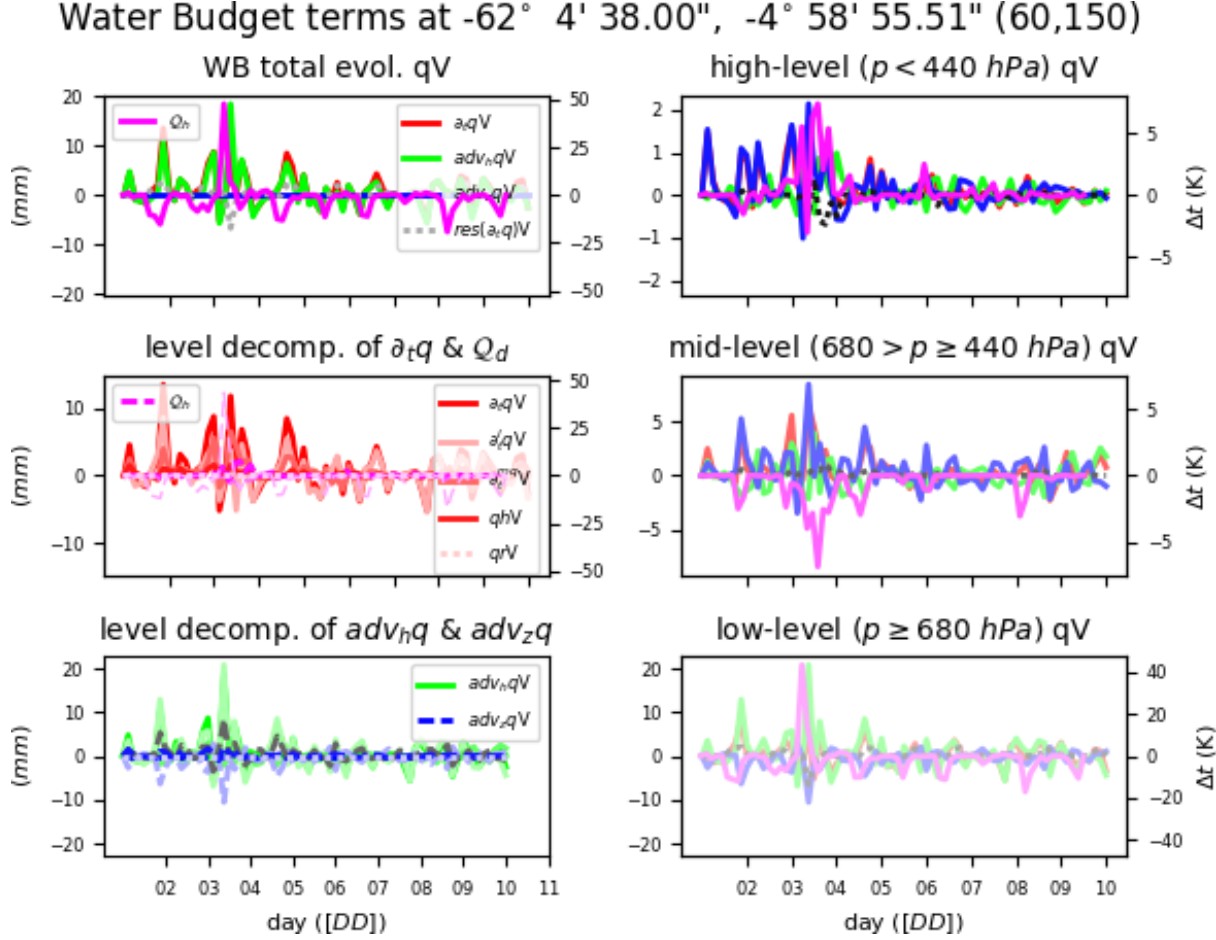

**Figure 12.** Water budget evolution at a given point for water vapour of vertically integrated water-budget terms: total tendency 'PW' ($\partial_t qv$, red), horizontal advection 'F' ($adv_h qv$, green), vertical advection 'Z' ($adv_z qv$, green), residual PW - F -Z ($res(\partial_t qv)$, gray dashed) and diabatic heating from micro-physics ($Q_d$, pink) (top left), only high-level vertically integrated values ($p < 440\ hPa$, top right), high/mid/low-level (degree of color intensity) decomposition of $partial_t qv$ (red) and $Q_d$ (pink) and their respective residuals as dashed lines (middle left), only mid-level vertically integrated values ($680 > p \leq 440\ hPa$, middle right), high/mid/low-level (degree of color intensity) decomposition of $adv_h qv$ (green) and $adv_z qv$ (blue) and their respective residuals as dashed lines (bottom left) and only low-level vertically integrated values ($p \geq 680\ hPa$, bottom right)





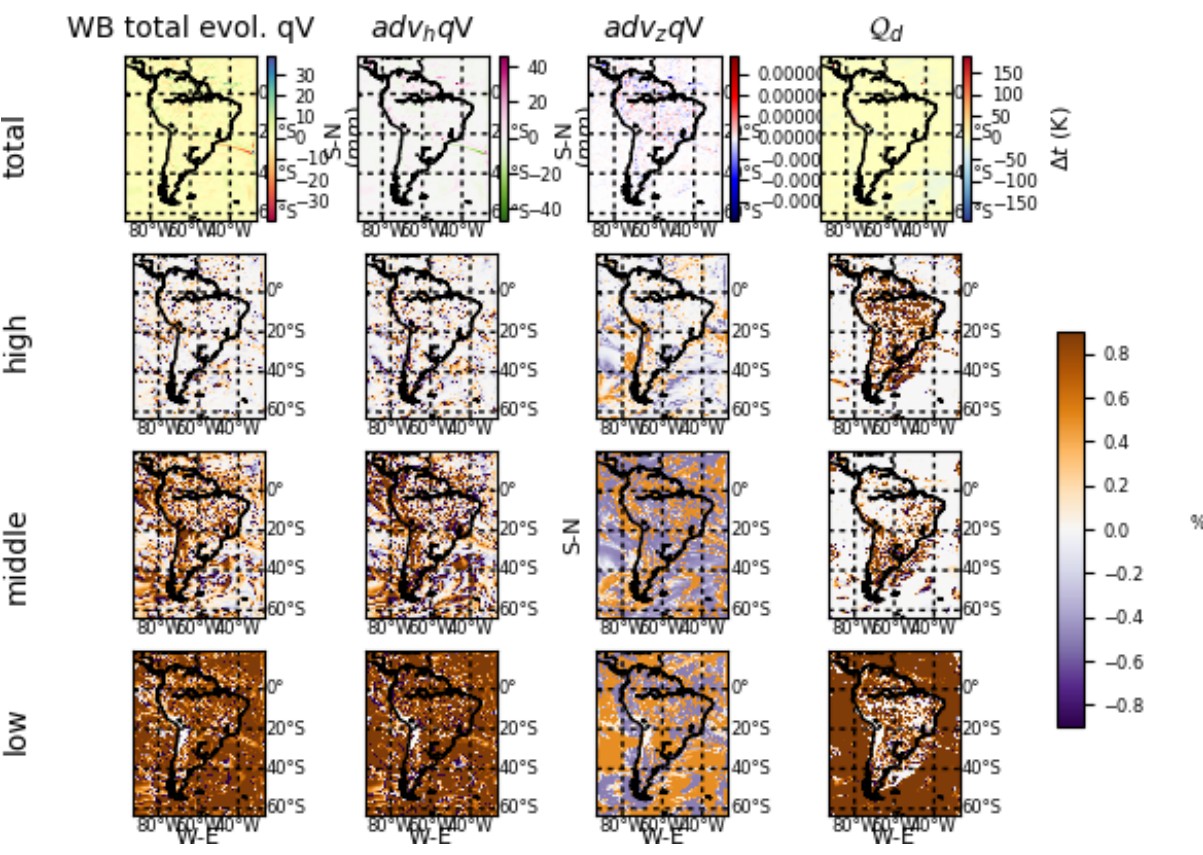

**Figure 13.** water vapour water budget maps of each component and diabatic heating from micro-physics at a given time and the percentual contribution at each different vertically integrated layer respective the total. total tendency 'PW' ($\partial_t qv$, first column), horizontal advection 'F' ($adv_h qv$, second col), vertical advection 'Z' ($adv_z qv$, third col.) and diabatic heating from micro-physics ($\mathcal{Q}_d$, 4th col). Percentage contribution of high level ($p < 440\ hPa$) integration to the total (second row), percentage for mid level ($680 > p \geq 440\ hPa$) integration to the total (third row) and percentage of low-level ($p \geq 680\ hPa$) integration (bottom row)





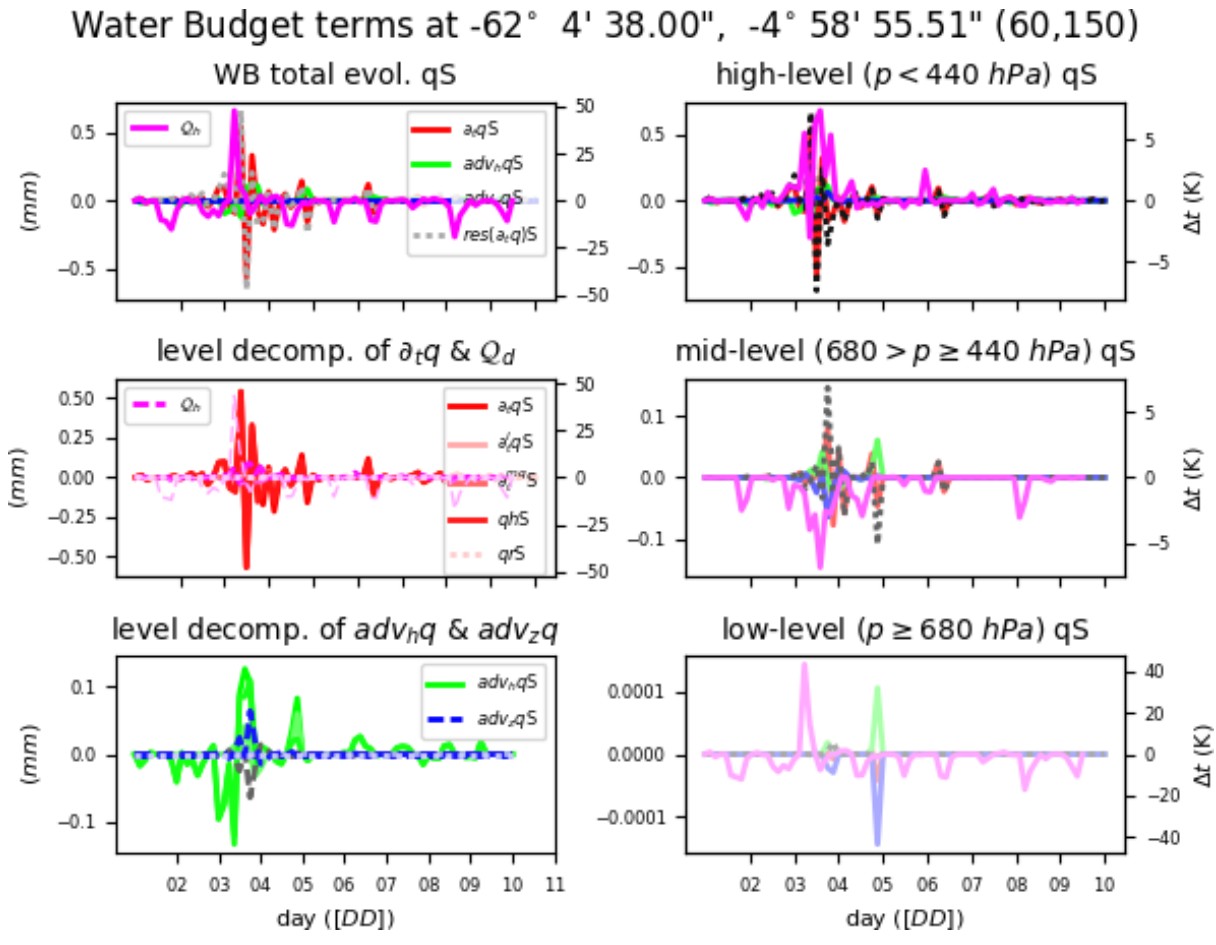

**Figure 14.** The same as in figure 12, but for snow

level present. Visibility is computed using equation 30 as in Kunkel (1984),

$$
fogvis = \begin{cases} visc = 0.027(QCLOUD \times 1000)^{-0.88} & QCLOUD \neq 0 \\ visi = 0.024(QICE \times 1000)^{-1.0} & QICE \neq 0 \end{cases}
$$

$$
fogvis = min(visc, visi)
$$

(30)

where $QCLOUD$: liquid water (cloud) mixing ratio ($kgkg^{-1}$), $QICE$: ice mixing ratio ($kgkg^{-1}$). Visibility values are
in $km$

– **RUC** [`visibility_diag = 2`]: Visibility is computed using relative humidity ($hur$) as implemented in the RUC
model (see equation 31 in Smirnova et al., 2000)

$$
fogvis = 60.0exp\left[\frac{-2.5(hur \times 100 - 15)}{80}\right]
$$

(31)



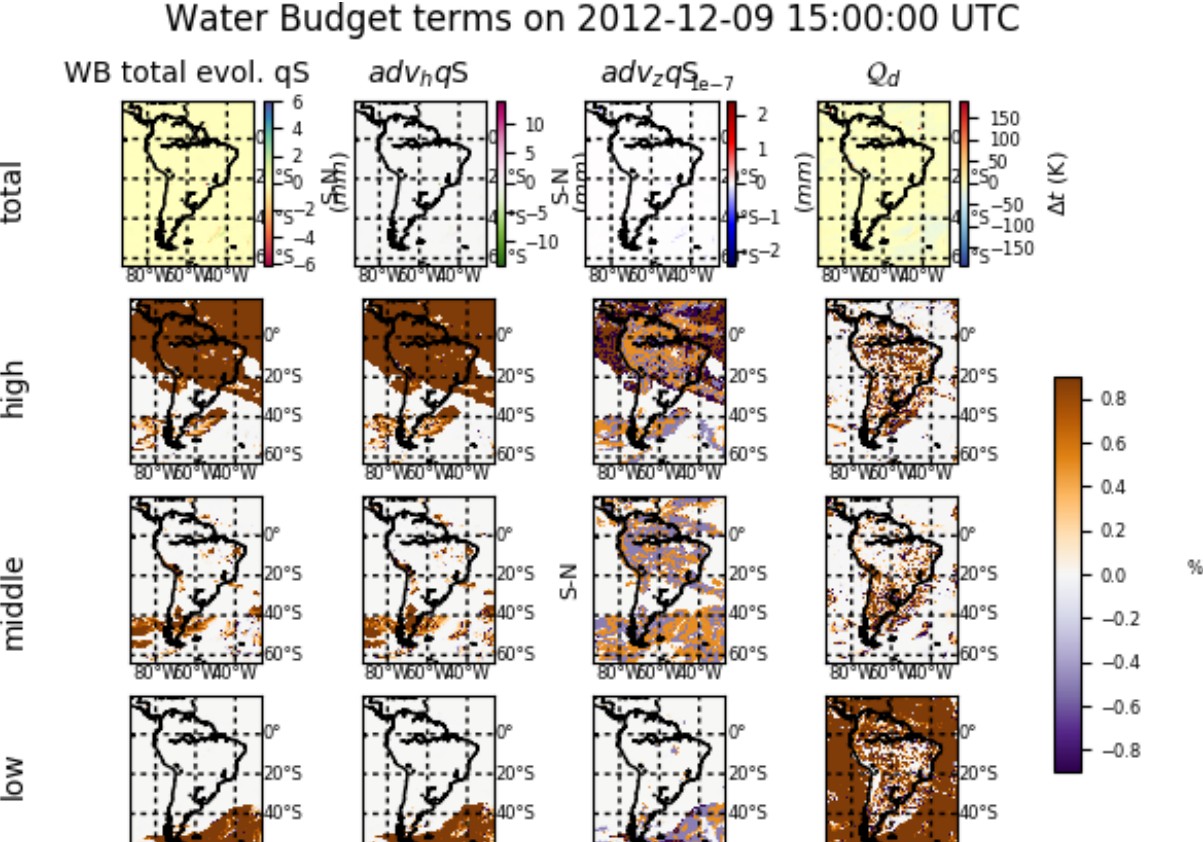

**Figure 15.** The same as in 13, bur for snow

where $hur$: relative humidity (1, previously computed) and can be from the 2-m diagnostics or the first model layer. Visibility values are in $km$

– **FRAM-L** [`visibility_diag = 3`], (default): Visibility is computed using relative humidity ($hur$) after (see equation 32 in Gultepe and Milbrandt, 2010). In this work, a probabilistic approach is proposed to compute the visibility in three different bins: 95% , 50% and 5% of probability to get certain visibility (for $rh > 30\%$). As a matter of compromise in the module, the calculation with the 50% of probability has been chosen as the preferred one. Therefore, this method provides the visibility that may occur with a 50% of probability.

$$fogvis^{prob} = \begin{cases} 95\% & -9.68 \times 10^{-14} hur^{7.19} + 52.20 \\ 50\% & -5.19 \times 10^{-10} hur^{5.44} + 40.10 \\ 5\% & -0.000114 rh^{2.70} + 27.45 \end{cases} \qquad (32)$$

where $hur$: relative humidity (1) and can be from 2-m diagnostics or first model layer. Visibility values are in $km$



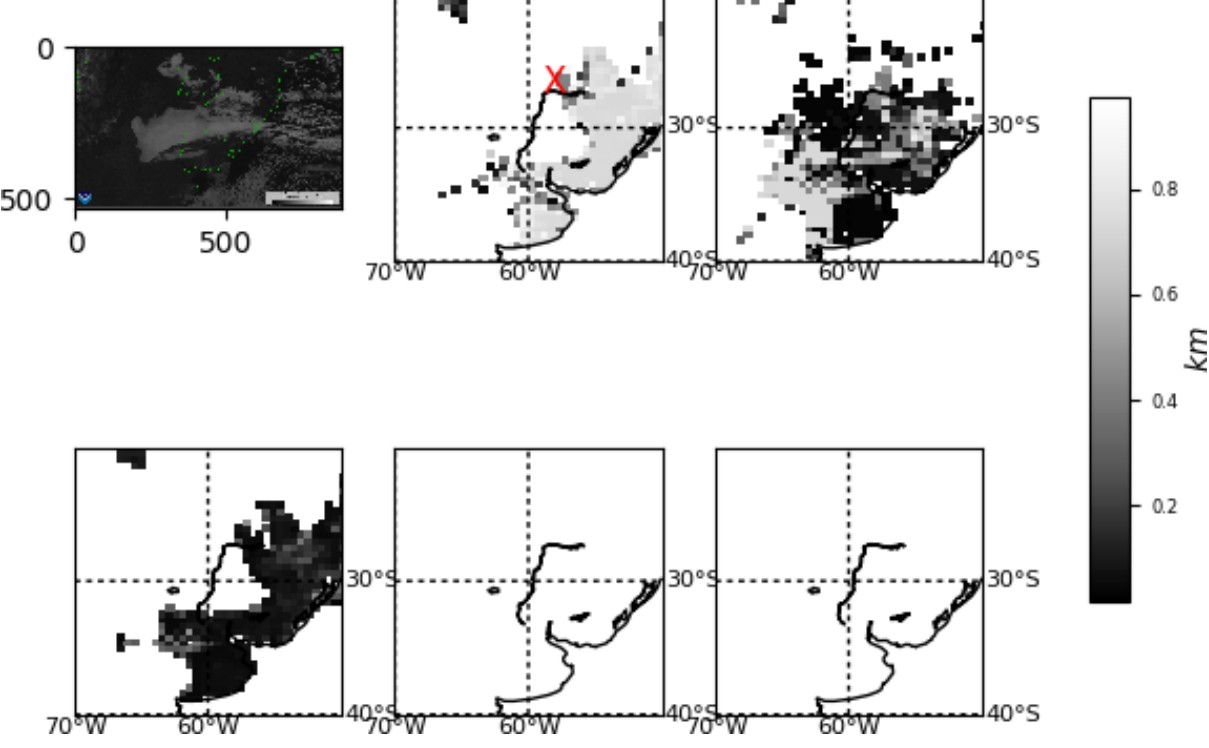

**Figure 16.** On 2007 June 30th at 12, comparison of the different configurations of the diagnostics of the mean fog visibility (in 1 hour) to the satellite image from GOES-12 (top left) at the same time in the visible channel (courtesy of NOAA-CLASS), default vis3vars1 (fogvisibility=3, fogvars=1; top middle), vis3vars2 (top right), vis1vars1 (bottom left), vis2vars1 (bottom middle) and vis2vars2 (bottom right)

Provided values of visibility during a fog event are: the minimum, maximum and mean values within output time steps (9freq) when fog occurred. Different choices are controlled throughout namelist variables: `visibility_diag` method of visibility computation, `fogvars` source of the relative humidity to be used as input in the visibility method. A user can choose to use the relative humidity from the first model layer ($hur$) `fogvars=1` (default value) or from the 2-m diagnostics ($hurs$)

5  `fogvars=2`. Some preliminary results of an extreme fog episode in central Argentina are provided in figure 16. Results strongly differ among fog implementations. The best agreement with a satellite visible channel picture for a given time of the event is obtained when the default setting is used ('FRAM-L' method with 'hur' values as input).

It is known that certain methods for calculating visibility relay on numerical adjustments on certain observational data taken under certain circumstances and at specific places (e.g.: for FRAM-L adjusted values come from observations from a Canadian

10  airport). It would be desirable to provide a more generic "all places/purposes" approach (if possible). It is recommended to take this variable with a certain care.





**tfog: time of presence of fog**

Fog can be diagnosed when the visibility is lower than 1 km (WMO, 2010b). `tfog` accounts for the period during which the grid point has visibility lower than 1 km during 9freq (see equation 33)

$$tfog = \sum_{it}^{9freq} \delta t, \ [vis(it) \leq 1km] \tag{33}$$

where $vis$: visibility $(km)$ below 1 km. $\delta t$: model time step $(s)$

## 5    Optimization

Regional climate dynamical downscaling experiments like the ones under the scope of CORDEX require long continuous simulations. A crucial element in these experiments is the real time necessary for model integration. therfore, a series of tests were carried out in order to investigate the impact of the module activation on the real time required for the model integration.

First version of the module (v1.0) was known to introduce about 40% decrease in time step speed of integration (highly dependent on HPC, model configuration and domain specifications). In order to improve model performance when the module is activated, the module was upgraded to the version v1.1. Since this new version, a series of optimizations of the code and pre-compilation flags were activated, with which a user can choose the level of complexity and amount of variables to be calculated. Following this implementation (instead via regular WRF namelist options), two main goals were achieved: (1) the

amount of variables kept in memory during model execution was reduced and (2) the number of conditions (mainly avoiding IF statements) to be checked and calculations at each execution step of integration were reduced as well.

A domain as shown in figure 17 has been set-up to perform short runs (5 days) to check the changes in performance of the WRF model when the module is activated in its different possible configurations. In order to avoid non-homogeneous communication among the cluster nodes (which would affect the analysis), all the simulations were executed on a single

node and with the WRF model compiled with intel and GNU Fortran compilers. Tests are performed at the HPC 'Fram' (https://www.sigma2.no/Fram) from the Norwegian academic HPC infrastructure. Fram is based on Lenovo NeXtScale nx360, constituted by CPU types: Intel E5-2683v4 2.1 GHz and Intel E7-4850v4 2.1 GHz.

The execution time is calculated as the mean elapsed time used during the 5-day model integration. Elapsed time necessary for each simulation step is available from the standard output of the model run (`rsl.error.0000` WRF ASCII file). In this

file, WRF users can get the elapsed time for all the time steps of the model and simulation doamins (see in figure 18). Different peaks of slower time steps coincide with Input/Output file operations, difference between day and night regimes and when different physical schemes (mainly the radiative scheme) are activated on a given frequency (namelist parameter `radt`). For a simulation covering 5-days with a time steps of 15 seconds, one obtains 28800 time steps. The sample of 28800 time steps is considered to be large enough to be representative for the mean time-step of the whole simulation.

Table 4 describes the different configurations and namelist options used in this performance test. The first simulation (labeled V381orig), which is used as reference, is the simulation with the original version of the model (here version 3.8.1) without





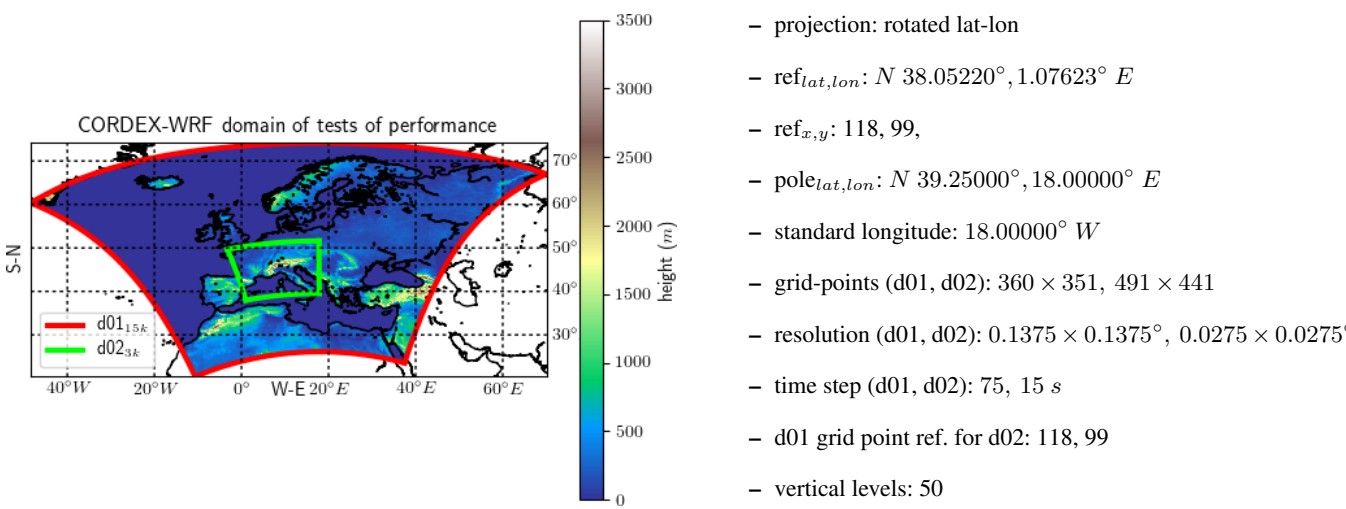

- projection: rotated lat-lon

- $\text{ref}_{lat,lon}$: $N\ 38.05220°, 1.07623°\ E$

- $\text{ref}_{x,y}$: 118, 99,

- $\text{pole}_{lat,lon}$: $N\ 39.25000°, 18.00000°\ E$

- standard longitude: $18.00000°\ W$

- grid-points (d01, d02): $360 \times 351,\ 491 \times 441$

- resolution (d01, d02): $0.1375 \times 0.1375°,\ 0.0275 \times 0.0275°$

- time step (d01, d02): 75, 15 $s$

- d01 grid point ref. for d02: 118, 99

- vertical levels: 50

**Figure 17.** 2-nested domain WRF3.8.1 configuration where different performance tests were carried out.

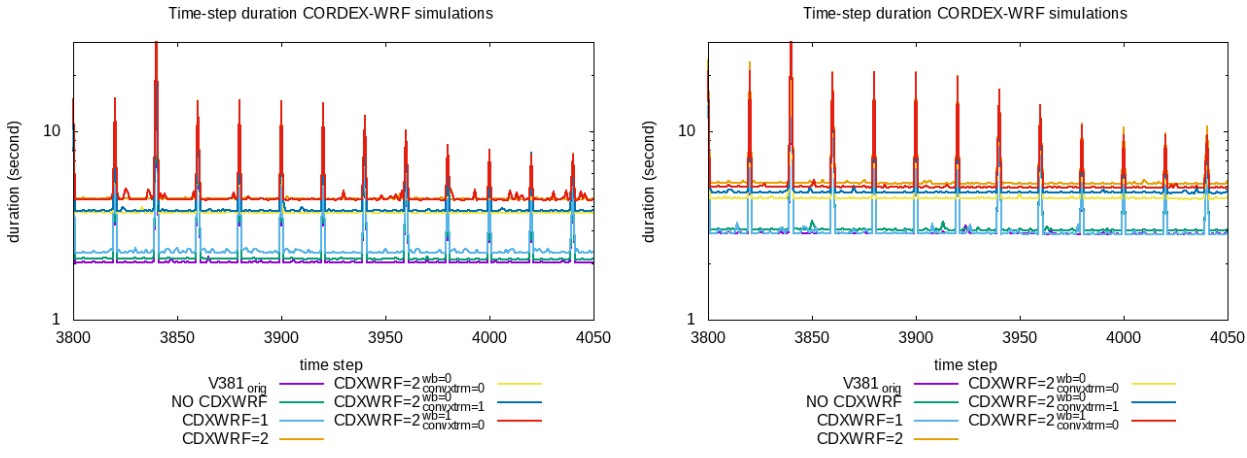

**Figure 18.** Elapsed times for each individual time step integration on nested domain d02 (time steps from number 3800 [simulating date 2014-01-01 15:35:00 UTC] to 4050 [2014-01-01 16:37:30 UTC]). Model was ran with different module configurations. See text for more details. Larger time steps are related to activation of the short/long-wave radiation scheme (every 5 minutes). For WRF compilation using 'ifort' (left) and 'gcc' compilers (right)



**Table 4.** Mean elapsed time step of 5-day simulation and difference in time with respect original version of the code (v381orig) for different model configurations. See text for more details

|  |  | ifort | | gcc | |
| --- | --- | --- | --- | --- | --- |
| label | description | $<t_{step}>$ (s) | diff. (%) | $<t_{step}>$ (s) | diff. (%) |
| v381orig | original WRF 3.8.1 | 2.4248 | - | 3.5174 | - |
| NOCDXWRF | without CDXWRF | 2.5058 | 3.34 | 3.6486 | 3.73 |
| CDXWRF1 | CDXWRF=1 | 2.6938 | 11.09 | 3.5070 | -0.27 |
| CDXWRF2 | CDXWRF=2 | 4.8296 | 99.17 | 5.9958 | 70.46 |
| CDXWRF2_00 | CDXWRF=2 wb_diag=0 & convxtrm_diag=0 | 4.2038 | 73.37 | 5.0736 | 44.24 |
| CDXWRF2_01 | CDXWRF=2 wb_diag=0 & convxtrm_diag=1 | 4.2388 | 74.81 | 5.4120 | 53.86 |
| CDXWRF2_10 | CDXWRF=2 wb_diag=1 & convxtrm_diag=0 | 4.8510 | 100.06 | 5.7534 | 63.57 |

the module. The other simulations are: activation of the module (setting `CORDEXDIAG`) without setting of the pre-compilation parameter `CDXWRF` (labeled NOCDXWRF); with pre-compilation parameter `CDXWRF=1` (CDXWRF1); with pre-compilation parameter `CDXWRF=2` and computing all extra calculations (CDXWRF2). Finally three more simulations (with `CDXWRF=2`) are made: without any extra calculation (CDXWRF2_00), without calculation of extra water-budget terms (CDXWRF2_01)

and without extremes from convection indices (CDXWRF2_10). Results might present some inconsistencies due to the fact that certain computation of diagnostics depend on the stability at each grid point which might vary from run to run and work load of the HPC.

Results show that all the simulations (except CDXWRF1 with gcc) where the module has been activated are slower than the simulation with the original version of the code (v381orig, $<t>= 2.4248[ifort], 3.5174[gcc]\ s$). Simulation with ver-

10 sion 1.3 of the module without pre-compilation flag `CDXWRF` (NOCDXWRF,. $<t>= 2.5058, 3.6486\ s$) is the second fastest. Simulation becomes slower when all the extra calculations are performed (CDXWRF2 $<t>= 4.8296, 5.9958\ s$). The heaviest part of the module is related to the water budget computation (`wb_diag=1`), since when comparing to the simulation without extra calculations (CDXWRF2_00, $<t>= 4.2038, 5.0736\ s$) there is an increase of only about 1, 9% ($<t_{step}>^{ifort}_{CDXWRF2\_01} / <t_{step}>^{ifort}_{CDXWRF2\_00}, <t_{step}>^{gcc}_{CDXWRF2\_01} / <t_{step}>^{gcc}_{CDXWRF2\_00}$) of mean time step

when only statistics of extreme convective indices is activated (CDXWRF2_01, $<t>= 4.2388, 5.4120s$), and 27, 19% ($<t_{step}>_{CDXWRF2\_10} / <t_{step}>_{CDXWRF2\_00} |_{ifort,gcc}$) when only water-budget terms are included (CDXWRF2_10, $<t>= 4.8510, 5.7534\ s$). Reduction on time-step for CDXWRF1 with gcc must be related to a moment where HPC 'Fram' experiences a period of very low working load.

These results are not conclusive (it should be tested in oder HPC resources, domains and compilers), but they provide a

20 first insight on how the number of variables included during the integration (in the derived type `grid`) has an important effect on model performance by reducing/increasing the required amount of memory. CDXWRF1 and CDXWRF2_00 perform the




same amount of computations of diagnostics, but the mean time step in CDXWRF2_00 is almost doubled (because in the CDXWRF2_00 case all the additional variables are defined but not diagnosed).

## 6  Summary and outlook

A series of modifications have been introduced into the WRF model in order to make it more suitable for CORDEX experiments. These modifications include the creation of a new module for the model and the modification of certain sections of the code. With all this included, users are now able to retrieve directly from the WRF output all the 'Core' and almost all the 'Tier1' CORDEX variables. Moreover, a series of additional variables not required by CORDEX which may be of a high interest to the regional climate modelling community have been included as well.

A WRF user participating in a CORDEX regional climate experiment will strongly benefit when activating the module presented here since it avoids most of the complex and time consuming post-processing efforts necessary to obtain diagnostics required by CORDEX. Furthermore, the module makes use of the WRF online interpolation to pressure levels of certain 3-dimensional variables which is an expensive computational task. In addition, statistical values (like minimum, fluxes or accumulation) are computed using the actual values obtained during the model integration. Since the use of the module avoids the post-processing of the model output, there is no need to keep large amount of additional fields which extremely reduces the requirements of disk storage (roughly 50 % less of disk storage).

In order to reduce the impact of the module during model integration, set-up of the module is managed before the compilation of the model via pre-compilation flags. This reduces the requirements of memory since the calculation of only required variables is activated. Different tests showed that the model performance is significantly improved when the code and the variables are constrained to the required ones (and not all available included), and managed via pre-compilation flags (and via conditional statements from the model namelist as it is usually done in the WRF model).

The module also establishes a first attempt to provide and implement generic definitions of certain diagnostics directly from regular model fields. This specific effort is intended to provide diagnostics which would not depend on the model configuration or/and the model itself, which will make intercomparisons between models more accurate, generic and trustworthy. If there is also a certain coordination (e.g. within the CORDEX community) of the definition of these diagnostics, climate studies will beneficed. Common definitions of diagnostics in a coordinated way across different modeling platforms, will ensure robustness of model intercomparisons.

There are certain variables from CORDEX 'Tier1' protocol, which are not yet introduced but required. These are: (1) the daily maximum wind speed of gust at 100 m (wsgsmax100) which is not yet introduced because of lack of an appropriate method, and (2) intra-cloud, ground and total lightning flashes (ic_lightning, cg_lightning, tot_lightning) are not yet introduced even though a lightning scheme is already implemented in WRF. However this implementation is not appropriate for small domain patches used during parallelization. It has been found (L. Fita and M. Álvarez, CIMA personal communication) that lightning flashes rates become patchy when size of the parallel domain partitions are smaller than the actual extent of the cell. Lightning methods (Price and Rind, 1992; Wong et al., 2013) require to encompass the whole convective cell to analyze the





updraft within the cell. When patches do not cover the whole cell, different values from the same cell at each parallel fraction of the domain is provided from which spatial inconsistencies arise.

There are some additional features that would make the module more useful for the climate regional community, which are also required by the CORDEX specifications. However, in order to prioritize the production of a basic, full working, and useful

version of the module as soon as possible, following aspects are planned for the future iterations and updates:

1. Flexibility-enhancement: Capacity to provide a more flexible module easy to be maintained and updated with new requirements from CORDEX or WRF model updates

2. Introduce new variables with a special focus on the implementation of more 'generic' variable definitions

3. CF-compliant/cmorzization output: WRF output does not fully follows CF-conventions. Thus a huge coding effort needs

to be done in order to provide a full CF-compliant output directly from it. A user still needs to process the output of the model in order to provide data following all the CORDEX guidelines. Due to uncovered steps of the CF-standard, a user of the WRF model still needs to: concatenate files, change names and attributes of variables, calculate temporal statistics over different periods (daily, monthly, seasonal) and provide the right time-variables in order to fully reach the CF-standard which followed by CORDEX. However, these steps are computationally lighter and easier to perform in

comparison to the computation of the different diagnostics and the vertical pressure interpolation already introduced in the module.

The incompatibility between WRF output and CF-convention can be overcame with the use of a complementary dedicated I/O library. This has been done for example in the RegIPSL platform (https://sourcesup.renater.fr/wiki/morcemed/Home, which uses WRF as atmospheric model) which uses XIOS (http://forge.ipsl.jussieu.fr/ioserver) libraries to manage the I/O.

*Code availability.* The discussed version of the module (1.3) in the present article for 4 different WRF versions (3.7.1 Fita, 2018a), (3.8.1 Fita, 2018b), (3.9.1.1 Fita, 2018c), (4.0 Fita, 2018d) is available throughout Digital Object Identificators (DOI):

– WRF-CORDEX module version 1.3 for WRFV3.7.1: DOI 10.5281/zenodo.1469639 (e.g. https://doi.org/10.5281/zenodo.1469639)

– WRF-CORDEX module version 1.3 for WRFV3.8.1: DOI 10.5281/zenodo.1469645

– WRF-CORDEX module version 1.3 for WRFV3.9.1.1:DOI 10.5281/zenodo.1469647

– WRF-CORDEX module version 1.3 for WRFV4.0: DOI 10.5281/zenodo.1469651

It is strongly recommended to make contact to lluis.fita@cima.fcen.uba.ar. This facilitates the creating of a community of users of the module, and make it more easy to share the updates and improvements with the users.

A WIKI page has been set-up in order to provide detailed instructions how to manage the module, and to give information about upcoming module versions and improvements. The page will be regularly updated and can be found at:

http://wiki.cima.fcen.uba.ar/mediawiki/index.php/CDXWRF





## Appendix B: CORDEX variables in WRF

### B1   Core variables

The variables given in Tables B1 and B2 are always provided when the module is activated with the use of the precompilation flag `CORDEXDIAG`. These variables appear in the auxiliary output file number 9. It is recommended to set the name of this file
as `wrfcdx_d<domain>_<date>`. Note that some variables might not be produced according to chosen namelist options.

### B2   Tier 1

The variables given in Table B3 are provided if the preprocessing variable `CDXWRF` is set to 1. These variables also appear in the auxiliary output file number 9. Note that some variables might not be produced according to chosen namelist options.

### B3   Additional

The variables from Tables from B4 to B7 are provided if the preprocessing variable `CDXWRF` is set to 2. These variables also appear in the auxiliary output file number 9. Note that some variables might not be produced according to chosen namelist options.

### B4   Instantaneous

The variables given in Table B8 are provided if some modifications are done in the WRF code. These variables represent
instantaneous values for certain number of variables which are internally used. These variables also appear in the auxiliary output file number 9. Note that some variables might not be produced according to chosen namelist options.





**Table B1.** Description of CORDEX Core variables provided with the module. kind specifies when the variable is computed; a: computed all time steps, o: only at output time, t: according to a frequency in the namelist, s: statistic value from internal integration values and initialized after each output time step

| CF name | WRF name | description | units | kind |
|---|---|---|---|---|
| 2D | | | | |
| lon | LON | LONGITUDE | degrees_east | o |
| lat | LAT | LATITUDE | degrees_north | o |
| cltmean | CLTMEAN | MEAN TOTAL CLOUDINESS IN CORDEX OUTPUT | % | t |
| cllmean | CLLMEAN | MEAN LOW-LEVEL CLOUDINESS ($p >= 68000$ Pa) IN CORDEX OUTPUT | % | t |
| clmmean | CLMMEAN | MEAN MID-LEVEL CLOUDINESS ($44000 <= p < 68000$ Pa) IN CORDEX OUTPUT | % | t |
| clhmean | CLHMEAN | MEAN HIGH-LEVEL CLOUDINESS ($p < 44000$ Pa) IN CORDEX OUTPUT | % | t |
| mrso | MRSO | TOTAL SOIL CONTENT | kgm-2 | o |
| prw | PRW | WATER VAPOR PATH | kgm-2 | o |
| psl | PSL | SEA LEVEL PRESSURE | Pa | o |
| clwvi | CLWVI | LIQUID WATER PATH | kgm-2 | o |
| clivi | CLIVI | ICE WATER PATH | kgm-2 | o |
| hurs[a] | HURS | 2M RELATIVE HUMIDITY | 1 | o |
| huss | HUSS | 2M SPECIFIC HUMIDITY | 1 | o |
| slw | SLW | TOTAL SOIL LIQUID WATER CONTENT | kgm-2 | o |
| uas | UAS | 10M EASTWARD WIND SPEED | ms-1 | o |
| vas | VAS | 10M NORTHWARD WIND SPEED | ms-1 | o |
| wsgsmax | WSGSMAX | Maximum near-surface wind speed of gust | ms-1 | s |
| usgsmax | USGSMAX | Eastward maximum near-surface wind speed of gust | ms-1 | s |
| vsgsmax | VSGSMAX | Northward maximum near-surface wind speed of gust | ms-1 | s |
| totwsgsmax | TOTWSGSMAX | Total (TKE + h. pr) Maximum near-surface wind speed of gust | ms-1 | s |
| totugsmax | TOTUGSMAX | Total Eastward maximum near-surface wind speed of gust | ms-1 | s |
| totvgsmax | TOTVGSMAX | Total Northward maximum near-surface wind speed of gust | ms-1 | s |
| wsz100max | WSZ100MAX | Maximum 100m nwind speed | ms-1 | s |
| uz100max | UZ100MAX | Eastward maximum 100 m wind speed | ms-1 | s |
| vz100max | VZ100MAX | Northward maximum 100 m wind speed | ms-1 | s |
| sund | SUND | SUNSHINE LENGTH (ac. time SWDOWN > 120. Wm-2) | second | s |

[a]needed for other variables





**Table B2.** Continuation of Table B1 of Core variables

| tauu | TAUU | northward downward wind stress at 10 m | m2s-2 | o |
|---|---|---|---|---|
| tauv | TAUV | eastward downward wind stress at 10 m | m2s-2 | o |
| tauugen | TAUUGEN | generic eastward downward wind stress at 10 m | m2s-2 | o |
| tauvgen | TAUVGEN | generic northward downward wind stress at 10 m | m2s-2 | o |
| rsds | RSDS | mean surface Downwelling Shortwave Radiation | Wm-2 | s |
| rlds | RLDS | mean surface Downwelling Longwave Radiation | Wm-2 | s |
| hfls | HFLS | mean surface Upward Latent Heat Flux | Wm-2 | s |
| hfss | HFSS | mean surface Upward Sensible Heat Flux | Wm-2 | s |
| rsus | RSUS | mean surface Upwelling Shortwave Radiation | Wm-2 | s |
| rlus | RLUS | mean surface Upwelling Longwave Radiation | Wm-2 | s |
| rsusgen | RSUSGEN | mean generic surface Upwelling Shortwave Radiation | Wm-2 | s |
| rlusgen | RLUSGEN | mean generic surface Upwelling Longwave Radiation | Wm-2 | s |
| evspsbl | EVSPSBL | mean evaporation | kgm-2s-1 | s |
| evspsblpot | EVSPSBLPOT | mean potential evapotranspiration | kgm-2s-1 | s |
| evspsblpotgen | EVSPSBLPOTGEN | mean generic potential evapotranspiration | kgm-2s-1 | s |
| cd | CDCDX | drag coefficient | - | o |
| cdgen | cdgen | generic drag coefficient | - | o |
| snc | SNC | mean snow area fraction | % | s |
| snd | SND | mean snow depth | m | s |
| mrros[a] | MRROS | mean surface Runoff | kgm-2s-1 | s |
| mrro[a] | MRRO | mean total Runoff | kgm-2s-1 | s |
| mrsol[a] | MRSOL | mean total water content of soil layer | kgm-2 | s |
| pr | PR | precipitation flux | kgm-2s-1 | s |
| prl | PRL | large scale precipitation flux | kgm-2s-1 | s |
| prc | PRC | convective precipitation flux | kgm-2s-1 | s |
| prsh | PRSH | shallow-cumulus precipitation flux | kgm-2s-1 | s |
| prsn | PRSN | solid precipitation flux | kgm-2s-1 | s |
| snw | SNW | accumulated snow precipitation | kgm-2 | s |
| rsdt | RSDT | mean top of the atmosphere (TOA) incident shortwave radiation | kgm-2 | s |
| rsut | RSUT | mean TOA outgoing shortwave radiation | kgm-2 | s |
| rlut | RLUT | mean TOA outgoing Longwave radiation | kgm-2 | s |
| ps | CDXPS | surface pressure | Pa | o |
| ts | CDXTS | skin temperature | K | o |

[a]unmasked to sea points



**Table B3.** As in B1, but for the description of CORDEX Tier1 variables provided with the module

| CF name | WRF name | description | units | kind |
|---|---|---|---|---|
| clgvi | CLGVI | GRAUPEL WATER PATH | kgm-2 | o |
| clhvi | CLHVI | HAIL WATER PATH | kgm-2 | o |
| zmlagen | ZMLAGEN | Generic boundary layer height theta(zmlagen) > min(theta[mix. layer]) + 1.5K | m | o |
| capemin[a] | CAPEMIN | MINIMUM CONVECTIVE AVAILABLE POTENTIAL ENERGY | Jkg-1 | s |
| capemax[a] | CAPEMAX | MAXIMUM CONVECTIVE AVAILABLE POTENTIAL ENERGY | Jkg-1 | s |
| capemean[a] | CAPEMEAN | MEAN CONVECTIVE AVAILABLE POTENTIAL ENERGY | Jkg-1 | s |
| cinmin[a] | CINMIN | MINIMUM CONVECTIVE INHIBITION | Jkg-1 | s |
| cinmax[a] | CINMAX | MAXIMUM CONVECTIVE INHIBITION | Jkg-1 | s |
| cinmeann[a] | CINMEAN | MEAN CONVECTIVE INHIBITION | Jkg-1 | s |
| lfcpmin[a] | LFCPMIN | MINIMUM PRESSURE LEVEL FREE CONVECTION | Pa | s |
| lfcpmax[a] | LFCPMAX | MAXIMUM PRESSURE LEVEL FREE CONVECTION | Pa | s |
| lfcpmean[a] | LFCPMEAN | MEAN PRESSURE LEVEL FREE CONVECTION | Pa | s |
| lfczmin[a] | LFCZMIN | MINIMUM HEIGHT LEVEL FREE CONVECTION | m | s |
| lfczmax[a] | LFCZMAX | MAXIMUM HEIGHT LEVEL FREE CONVECTION | m | s |
| lfczmean[a] | LFCZMEAN | MEAN HEIGHT LEVEL FREE CONVECTION | m | s |
| limin[a] | LIMIN | MINIMUM LIFTED INDEX | 1 | s |
| limax[a] | LIMAX | MAXIMUM LIFTED INDEX | 1 | s |
| limean[a] | LIMEAN | MEAN LIFTED INDEX | 1 | s |

[a]it will be computed if namelist parameter `convxtrm_diag` is set to 1

*Acknowledgements.* All the coders of WRF, LMDZ and ORCHIDEE models are acknowledged for their hard work on the developing and maintaining of the models. M. A. Jiménez from Universitat de les Illes Balears is acknowledged by her explanations on certain PBL calculations. D. Argüeso from UIB. V. Galligani, J. Ruiz and M. Sebastián from CIMA are also acknowldeged by their commentaries. A. Sörensson and E. Borrell are also acknowledged by their assistance. Implementation tests where performed in CIMA HPC resources 'hydra'
5    cluster supported by the High Performance Computing National System (SNCAD) of Argentina L. Fita thanks the CIMA-IT support for their work. E. Katragkou and I. Sofiadis acknowledge the technical support and provision of resources from the Scientific Computing Center of AUTH (https://it.auth.gr/el) and the GRNET National HPC infrastucture (https://hpc.grnet.gr/). J.Milovac gratefully acknowledge the support by the German Science Foundation (DFG) through project FOR 1695 and the supercomputing center HLRS in Stuttgart Germany for granting the computing time necessary for the test simulations. T. Lorenz acknowledges the support of NOTUR project no. NN9280K and the Research
10   Council of Norway and its basic institute support of their strategic project on Climate Services. Figures were produced with python (except performarnce tests drawn with GNUplot) and L. Fita thanks the development of matplotlib above which he developed and make available





**Table B4.** As in B1, but for the description of additional variables provided with the module

| CF name | WRF name | description | units | kind |
|---------|----------|-------------|-------|------|
| tdsmin | TDSMIN | minimum surface dew point temperature | K | s |
| tdsmax | TDSMAX | maximum surface dew point temperature | K | s |
| tdsmean | TDSMEAN | mean surface dew point temperature | K | s |
| tfog | TFOG | time of presence of fog | seconds | s |
| fogvisbltymin | FOGVISBLTYMIN | minimum of visibility inside fog | km | s |
| fogvisbltymax | FOGVISBLTYMAX | maximum of visibility inside fog | km | s |
| fogvisbltymean | FOGVISBLTYMEAN | mean of visibility inside fog | km | s |
| **3D** | | | | |
| hur[a] | HUR | AIR RELATIVE HUMIDITY | 1 | a |
| hus | HUS | AIR SPECIFIC HUMIDITY | 1 | o |
| zg[a] | ZG | AIR GEOPOTENTIAL HEIGHT | m | a |
| press[a] | PRESS | AIR PRESSURE | Pa | a |
| ta[a] | TA | AIR TEMPERATURE | K | a |
| ua | UA | EARTH ROTATED AIR EASTWARD WIND SPEED | ms-1 | o |
| va | VA | EARTH ROTATED AIR NORTHWARD WIND SPEED | ms-1 | o |
| **Water-Budget[b]** | | | | |
| $Q_{hac}$ | WBACDIABH | Water Budget column integrated and time accumulation of diabatic heating from Micro-Physics | K | s |
| $\partial_t$qvac | WBACPW | Water Budget column integrated and time accumulated for water vapor content | mm | s |
| $\partial_t$qcac | WBACPWC | Water Budget col. int. & time accumulated for cloud content | mm | s |
| $\partial_t$qrac | WBACPWR | Water Budget col. int. & time accumulated for rain content | mm | s |
| $\partial_t$qsac | WBACPWS | Water Budget col. int. & time accumulated for snow content | mm | s |
| $\partial_t$qiac | WBACPWI | Water Budget col. int. & time accumulated for ice content | mm | s |
| $\partial_t$qhac | WBACPWH | Water Budget col. int. & time accumulated for hail content | mm | s |
| $\partial_t$qgac | WBACPWG | Water Budget col. int. & time accumulated for graupel content | mm | s |

[a]needed for other variables

[b]Variables will be computed if namelist parameter `output_wb` is set to 1




**Table B5.** Continuation of Table B4 of additional variables

| | | | | |
|---|---|---|---|---|
| $\mathrm{adv}_h\mathrm{qvac}$ | WBACF | W.B. c-int. acc. hor. convergence of water vapour (+, conv.; -, div.) | mm | s |
| $\mathrm{adv}_h\mathrm{qcac}$ | WBACFC | W.B. c-int. acc. hor. convergence of cloud (+, conv.; -, div.) | mm | s |
| $\mathrm{adv}_h\mathrm{qrac}$ | WBACFR | W.B. c-int. acc. hor. convergence of rain (+, conv.; -, div.) | mm | s |
| $\mathrm{adv}_h\mathrm{qsac}$ | WBACFS | W.B. c-int. acc. hor. convergence of snow (+, conv.; -, div.) | mm | s |
| $\mathrm{adv}_h\mathrm{qiac}$ | WBACFI | W.B. c-int. acc. hor. convergence of ice (+, conv.; -, div.) | mm | s |
| $\mathrm{adv}_h\mathrm{qhac}$ | WBACFH | W.B. c-int. acc. hor. convergence of hail (+, conv.; -, div.) | mm | s |
| $\mathrm{adv}_h\mathrm{qgac}$ | WBACFG | W.B. c-int. acc. hor. convergence of graupel (+, conv.; -, div.) | mm | s |
| $\mathrm{adv}_z\mathrm{qvac}$ | WBACZ | W.B. c-int. acc. ver. convergence of water vapour (+, conv.; -, div.), always 0 | mm | s |
| $\mathrm{adv}_z\mathrm{qcac}$ | WBACZC | W.B. c-int. acc. ver. convergence of cloud (+, conv.; -, div.) | mm | s |
| $\mathrm{adv}_z\mathrm{qrac}$ | WBACZR | W.B. c-int. acc. ver. convergence of rain (+, conv.; -, div.) | mm | s |
| $\mathrm{adv}_z\mathrm{qsac}$ | WBACZS | W.B. c-int. acc. ver. convergence of snow (+, conv.; -, div.) | mm | s |
| $\mathrm{adv}_z\mathrm{qiac}$ | WBACZI | W.B. c-int. acc. ver. convergence of ice (+, conv.; -, div.) | mm | s |
| $\mathrm{adv}_z\mathrm{qhac}$ | WBACZH | W.B. c-int. acc. ver. convergence of hail (+, conv.; -, div.) | mm | s |
| $\mathrm{adv}_z\mathrm{qgac}$ | WBACZG | W.B. c-int. acc. ver. convergence of graupel (+, conv.; -, div.) | mm | s |
| $Q_h^l\mathrm{ac}$ | WBACDIABHL | W.B. low level acc. of diabatic heating from MP | K | s |
| $Q_h^m\mathrm{ac}$ | WBACDIABHM | W.B. mid-level acc. of diabatic heating from MP | K | s |
| $Q_h^m\mathrm{ac}$ | WBACDIABHH | W.B. high-level acc. of diabatic heating from MP | K | s |
| $\partial_t^l\mathrm{qvac}$ | WBACPWLV | W.B. low level (p >=68000 Pa) acc. for QV | mm | s |
| $\partial_t^m\mathrm{qvac}$ | WBACPWMV | W.B. mid level (44000 Pa <=p <68000 Pa) acc. for QV | mm | s |
| $\partial_t^h\mathrm{qvac}$ | WBACPWHV | W.B. high level (p <44000 Pa) acc. for QV | mm | s |
| $\mathrm{adv}_h^l\mathrm{qvac}$ | WBACFLV | W.B. low-lev. acc. hor. convergence of QV | mm | s |
| $\mathrm{adv}_h^m\mathrm{qvac}$ | WBACFMV | W.B. mid-lev. acc. hor. convergence of QV | mm | s |
| $\mathrm{adv}_h^h\mathrm{qvac}$ | WBACFHV | W.B. high-lev. acc. hor. convergence of QV | mm | s |
| $\mathrm{adv}_z^l\mathrm{qvac}$ | WBACZLV | W.B. low level acc. ver. convergence of QV | mm | s |
| $\mathrm{adv}_z^m\mathrm{qvac}$ | WBACZMV | W.B. mid level acc. ver. convergence of QV | mm | s |
| $\mathrm{adv}_z^h\mathrm{qvac}$ | WBACZHV | W.B. high level acc. ver. convergence of QV | mm | s |

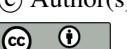



**Table B6.** 2nd continuation of Table B4 of additional variables

| | | | | |
|---|---|---|---|---|
| $\partial_t^l$qcac | WBACPWLC | W.B. low level (p >=68000 Pa) acc. for QC | mm | s |
| $\partial_t^m$qcac | WBACPWMC | W.B. mid level (44000 Pa <=p <68000 Pa) acc. for QC | mm | s |
| $\partial_t^h$qcac | WBACPWHC | W.B. high level (p <44000 Pa) acc. for QC | mm | s |
| $adv_h^l$qcac | WBACFLC | W.B. low-lev. acc. hor. convergence of QC | mm | s |
| $adv_h^m$qcac | WBACFMC | W.B. mid-lev. acc. hor. convergence of QC | mm | s |
| $adv_h^h$qcac | WBACFHC | W.B. high-lev. acc. hor. convergence of QC | mm | s |
| $adv_z^l$qcac | WBACZLC | W.B. low level acc. ver. convergence of QC | mm | s |
| $adv_z^m$qcac | WBACZMC | W.B. mid level acc. ver. convergence of QC | mm | s |
| $adv_z^h$qcac | WBACZHC | W.B. high level acc. ver. convergence of QC | mm | s |
| $\partial_t^l$qrac | WBACPWLR | W.B. low level (p >=68000 Pa) acc. for QR | mm | s |
| $\partial_t^m$qrac | WBACPWMR | W.B. mid level (44000 Pa <=p <68000 Pa) acc. for QR | mm | s |
| $\partial_t^h$qrac | WBACPWHR | W.B. high level (p <44000 Pa) acc. for QR | mm | s |
| $adv_h^l$qrac | WBACFLR | W.B. low-lev. acc. hor. convergence of QR | mm | s |
| $adv_h^m$qrac | WBACFMR | W.B. mid-lev. acc. hor. convergence of QR | mm | s |
| $adv_h^h$qrac | WBACFHR | W.B. high-lev. acc. hor. convergence of QR | mm | s |
| $adv_z^l$qrac | WBACZLR | W.B. low level acc. ver. convergence of QR | mm | s |
| $adv_z^m$qrac | WBACZMR | W.B. mid level acc. ver. convergence of QR | mm | s |
| $adv_z^h$qrac | WBACZHR | W.B. high level acc. ver. convergence of QR | mm | s |
| $\partial_t^l$qsac | WBACPWLS | W.B. low level (p >=68000 Pa) acc. for QS | mm | s |
| $\partial_t^m$qsac | WBACPWMS | W.B. mid level (44000 Pa <=p <68000 Pa) acc. for QS | mm | s |
| $\partial_t^h$qsac | WBACPWHS | W.B. high level (p <44000 Pa) acc. for QS | mm | s |
| $adv_h^l$qsac | WBACFLS | W.B. low-lev. acc. hor. convergence of QS | mm | s |
| $adv_h^m$qsac | WBACFMS | W.B. mid-lev. acc. hor. convergence of QS | mm | s |
| $adv_h^h$qsac | WBACFHS | W.B. high-lev. acc. hor. convergence of QS | mm | s |
| $adv_z^l$qsac | WBACZLS | W.B. low level acc. ver. convergence of QS | mm | s |
| $adv_z^m$qsac | WBACZMS | W.B. mid level acc. ver. convergence of QS | mm | s |
| $adv_z^h$qsac | WBACZHS | W.B. high level acc. ver. convergence of QS | mm | s |
| $\partial_t^l$qiac | WBACPWLI | W.B. low level (p >=68000 Pa) acc. for QI | mm | s |
| $\partial_t^m$qiac | WBACPWMI | W.B. mid level (44000 Pa <=p <68000 Pa) acc. for QI | mm | s |
| $\partial_t^h$qiac | WBACPWHI | W.B. high level (p <44000 Pa) acc. for QI | mm | s |
| $adv_h^l$qiac | WBACFLI | W.B. low-lev. acc. hor. convergence of QI | mm | s |
| $adv_h^m$qiac | WBACFMI | W.B. mid-lev. acc. hor. convergence of QI | mm | s |
| $adv_h^h$qiac | WBACFHI | W.B. high-lev. acc. hor. convergence of QI | mm | s |
| $adv_z^l$qiac | WBACZLI | W.B. low level acc. ver. convergence of QI | mm | s |
| $adv_z^m$qiac | WBACZMI | W.B. mid level acc. ver. convergence of QI | mm | s |
| $adv_z^h$qiac | WBACZHI | W.B. high level acc. ver. convergence of QI | mm | s |



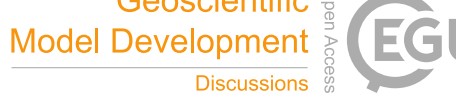

**Table B7.** 2nd continuation of Table B4 of additional variables

| | | | | |
|---|---|---|---|---|
| $\partial_t^l$qgac | WBACPWLG | W.B. low level (p >=68000 Pa) acc. for QG | mm | s |
| $\partial_t^m$qgac | WBACPWMG | W.B. mid level (44000 Pa <=p <68000 Pa) acc. for QG | mm | s |
| $\partial_t^h$qgac | WBACPWHG | W.B. high level (p <44000 Pa) acc. for QG | mm | s |
| $adv_h^l$qgac | WBACFLG | W.B. low-lev. acc. hor. convergence of QG | mm | s |
| $adv_h^m$qgac | WBACFMG | W.B. mid-lev. acc. hor. convergence of QG | mm | s |
| $adv_h^h$qgac | WBACFHG | W.B. high-lev. acc. hor. convergence of QG | mm | s |
| $adv_z^l$qgac | WBACZLG | W.B. low level acc. ver. convergence of QG | mm | s |
| $adv_z^m$qgac | WBACZMG | W.B. mid level acc. ver. convergence of QG | mm | s |
| $adv_z^h$qgac | WBACZHG | W.B. high level acc. ver. convergence of QG | mm | s |
| $\partial_t^l$qhac | WBACPWLH | W.B. low level (p >=68000 Pa) acc. for QH | mm | s |
| $\partial_t^m$qhac | WBACPWMH | W.B. mid level (44000 Pa <=p <68000 Pa) acc. for QH | mm | s |
| $\partial_t^h$qhac | WBACPWHH | W.B. high level (p <44000 Pa) acc. for QH | mm | s |
| $adv_h^l$qhac | WBACFLH | W.B. low-lev. acc. hor. convergence of QH | mm | s |
| $adv_h^m$qhac | WBACFMH | W.B. mid-lev. acc. hor. convergence of QH | mm | s |
| $adv_h^h$qhac | WBACFHH | W.B. high-lev. acc. hor. convergence of QH | mm | s |
| $adv_z^l$qhac | WBACZLH | W.B. low level acc. ver. convergence of QH | mm | s |
| $adv_z^m$qhac | WBACZMH | W.B. mid level acc. ver. convergence of QH | mm | s |
| $adv_z^h$qhac | WBACZHH | W.B. high level acc. ver. convergence of QH | mm | s |

a suite in python for netCDF management and plotting purposes called 'PyNCplot' (available from http://www.xn--llusfb-5va.cat/python/PyNCplot). Authors thank the commentaries of the topical editor which remarkebly improve the manuscipt.




**Table B8.** As in B1, but for the description of CORDEX 'instantaneous' variables provided with the module

| CF name | WRF name | description | units | kind |
|---|---|---|---|---|
| clt | CLT | TOTAL CLOUDINESS | 1 | t |
| cll | CLL | LOW-LEVEL CLOUDINESS (p >= 68000 Pa) | 1 | t |
| clm | CLM | MID-LEVEL CLOUDINESS (44000 <= p < 68000 Pa) | 1 | t |
| clh | CLH | HIGH-LEVEL CLOUDINESS (p < 44000 Pa) | 1 | t |
| cape | CDXCAPE | CONVECTIVE AVAILABLE POTENTIAL ENERGY | Jkg-1 | a/o[a] |
| cin | CIN | CONVECTIVE INHIBITION | Jkg-1 | a/o[a] |
| lfcp | LFCP | PRESSURE LEVEL FREE CONVECTION | Pa | a/o[a] |
| lfcz | LFCZ | HEIGHT LEVEL FREE CONVECTION | m | a/o[a] |
| li | LI | LIFTED INDEX | 1 | a/o[a] |
| wsgs | WSGS | near-surface wind speed of gust | ms-1 | a |
| usgs | USGS | Eastward near-surface wind speed of gust | ms-1 | a |
| vsgs | VSGS | Northward near-surface wind speed of gust | ms-1 | a |
| wsgspercen | WSGSPERCEN | Percentage of time steps where grid point got wind gust | % | s |
| totwsgspercen | TOTWSGSPERCEN | Percentage of time steps where grid point got total wind gust | % | s |
| wsz100 | WSZ100 | 100m wind speed | ms-1 | a |
| uz100 | UZ100 | Eastward 100 m wind speed | ms-1 | a |
| vz100 | VZ100 | Northward 100 m wind speed | ms-1 | a |
| fog | FOG | Whether there is fog (1: yes [vis < 1km]; 0: not) | - | a |
| fogvisblty | FOGVISBLTY | visibility inside fog | km | a |
| tds | TDS | surface dew point temperature | K | a |

[a]depending on namelist parameter `convxtrm_diag` 0:o, 1:a

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



**Table B9.** Continuation of Table B8

| | 3D-Water budget | | | |
|---|---|---|---|---|
| $qv_t tend$ | QVTTEND | inter time step water vapor tendency | kgkg-1s-1 | a |
| $qc_t tend$ | QCTTEND | inter time step cloud tendency | kgkg-1s-1 | a |
| $qr_t tend$ | QRTTEND | inter time step rain tendency | kgkg-1s-1 | a |
| $qs_t tend$ | QSTTEND | inter time step snow tendency | kgkg-1s-1 | a |
| $qi_t tend$ | QITTEND | inter time step ice tendency | kgkg-1s-1 | a |
| $qh_t tend$ | QHTTEND | inter time step hail tendency | kgkg-1s-1 | a |
| $qg_t tend$ | QGTTEND | inter time step graupel tendency | kgkg-1s-1 | a |
| $qv_h adv$ | QV_HADV | Instantaneous QV Horizontal advection | kgkg-1 | a |
| $qc_h adv$ | QC_HADV | Instantaneous QC Horizontal advection | kgkg-1 | a |
| $qr_h adv$ | QR_HADV | Instantaneous QR Horizontal advection | kgkg-1 | a |
| $qs_h adv$ | QS_HADV | Instantaneous QS Horizontal advection | kgkg-1 | a |
| $qi_h adv$ | QI_HADV | Instantaneous QI Horizontal advection | kgkg-1 | a |
| $qh_h adv$ | QH_HADV | Instantaneous QH Horizontal advection | kgkg-1 | a |
| $qg_h adv$ | QG_HADV | Instantaneous QG Horizontal advection | kgkg-1 | a |
| $qv_z adv$ | QV_ZADV | Instantaneous QV Vertical advection | kgkg-1 | a |
| $qc_z adv$ | QC_ZADV | Instantaneous QC Vertical advection | kgkg-1 | a |
| $qr_z adv$ | QR_ZADV | Instantaneous QR Vertical advection | kgkg-1 | a |
| $qs_z adv$ | QS_ZADV | Instantaneous QS Vertical advection | kgkg-1 | a |
| $qi_z adv$ | QI_ZADV | Instantaneous QI Vertical advection | kgkg-1 | a |
| $qh_z adv$ | QH_ZADV | Instantaneous QH Vertical advection | kgkg-1 | a |
| $qg_z adv$ | QG_ZADV | Instantaneous QG Vertical advection | kgkg-1 | a |

Brasseur, O.: Development and Application of a Physical Approach to Estimating Wind Gusts, Monthly Weather Review, 129, 5–25, https://doi.org/10.1175/1520-0493(2001)129<0005:DAAOAP>2.0.CO;2, https://DOI.org/10.1175/1520-0493(2001)129<0005:DAAOAP>2.0.CO;2, 2001.

Businger, J. A., Wyngaard, J. C., Izumi, Y., and Bradley, E. F.: Flux-Profile Relationships in the Atmospheric Surface Layer, J. Atmos. Sci., 28, 181–189, https://doi.org/10.1175/1520-0469(1971)028<0181:FPRITA>2.0.CO;2, https://doi.org/10.1175/1520-0469(1971)028<0181:FPRITA>2.0.CO;2, 1971.

Coppola, E., Sobolowski, S., Pichelli, E., Raffaele, F., Ahrens, B., Anders, I., Ban, N., Bastin, S., Belda, M., Belusic, D., Caldas-Alvarez, A., Cardoso, R. M., Davolio, S., Dobler, A., Fernández, J., Fita, L., Fumiere, Q., Giorgi, F., Goergen, K., Guettler, I., Halenka, T., Heinzeller, D., Hodnebrog, O., Jacob, D., Kartsios, S., Katragkou, E., Kendon, E., Khodayar, S., Kunstmann, H., Knist, S., Lavín, A., Lind, P., Lorenz, T., Maraun, D., Marelle, L., van Meijgaard, E., Milovac, J., Myhre, G., H.-J.Panitz, Piazza, M., Raffa, M., Raub, T., Rockel, B., Schär, C., Sieck, K., Soares, P. M. M., Somot, S., Srnec, L., Stocchi, P., Tölle, M., Truhetz, H., Vautard, R., de Vries, H., and Warrach-Sagi, K.: A





first-of-its-kind multi-model convection permitting ensemble for investigating convective phenomena over Europe and the Mediterranean, Clim. Dyn., under revision, 2018.

de Rosnay, P., Polcher, J., Bruen, M., and Laval, K.: Impact of a physically based soil water flow and soil-plant interaction representation for modeling large-scale land surface processes, J. Geophys. Res. (Atmospheres), 107(D11), https://doi.org/10.1029/2001JD000634, 2002.

Domínguez, M., Romera, R., Sánchez, E., Fita, L., Fernández, J., Jiménez-Guerrero, P., Montávez, J. P., Cabos, W. D., Liguori, G., and Gaertner, M. A.: Present climate precipitation and temperature extremes over Spain from a set of high resolution RCMs, Climate research, 58, 149–164, https://doi.org/10.3354/cr01186, 2013.

Evans, J. P., Ji, F., Lee, C., Smith, P., Argüeso, D., and Fita, L.: A regional climate modelling projection ensemble experiment - NARCliM, Geoscientific Model Development, 7(2), 621–629, https://doi.org/10.5194/gmd-7-621-2014, 2014.

Fita, L.: WRF-CORDEX module version 1.3 for WRFV3.7.1, https://doi.org/10.5281/zenodo.1469639, https://doi.org/10.5281/zenodo.1469639, 2018a.

Fita, L.: WRF-CORDEX module version 1.3 for WRFV3.8.1, https://doi.org/10.5281/zenodo.1469645, https://doi.org/10.5281/zenodo.1469645, 2018b.

Fita, L.: WRF-CORDEX module version 1.3 for WRFV3.9.1.1, https://doi.org/10.5281/zenodo.1469647, https://doi.org/10.5281/zenodo.1469647, 2018c.

Fita, L.: WRF-CORDEX module version 1.3 for WRFV4.0, https://doi.org/10.5281/zenodo.1469651, https://doi.org/10.5281/zenodo.1469651, 2018d.

Fita, L. and Flaounas, E.: Medicanes as subtropical cyclones: the December 2005 case from the perspective of surface pressure tendency diagnostics and atmospheric water budget, Q. J. Royal Meteo. Soc., 144, 1028–1044, https://doi.org/10.1002/qj.3273, qJ-17-0198.R2, 20  2018.

Fita, L., Fernández, J., and García-Díez, M.: CLWRF: WRF modifications for regional climate simulation under future scenarios, Proceedings of 11th WRF Users' Workshop, 2010.

Fu, C., Wang, S., Xiong, Z., Gutowski, W. J., Lee, D. K., McGregor, J. L., Sato, Y., Kato, H., Kim, J. W., and Suh, M. S.: Regional climate model intercomparison project for Asia, Bull. Amer. Meteor. Soc., 86(2), 257–266, 2005.

García-Díez, M., Fernández, J., Fita, L., and Yagüe, C.: Seasonal dependence of WRF model biases and sensitivity to PBL schemes over Europe, Q. J. of Roy. Met. Soc., 139, 501–514, https://doi.org/10.1002/qj.1976, 2013.

Garratt, J.: The Atmospheric Boundary Layer, Cambridge Univ. Press, Cambridge, U.K., 1992.

Geleyn, J. F. and Hollingsworth, A.: An economical analytical method for the computation of the interaction between scattering and line absorption of radiation, Contrib. Atmos. Phys, 52, 1–16, 1979.

Giorgi, F. and Gutowski, W. J.: Regional Dynamical Downscaling and the CORDEX Initiative, Annual Review of Environment and Resources, 40, 467–490, https://doi.org/10.1146/annurev-environ-102014-021217, 2015.

Giorgi, F. and Mearns, L. O.: Approaches to the simulation of regional climate change: A review, Reviews of Geophysics, 29, 191–216, https://doi.org/10.1029/90RG02636, http://dx.doi.org/10.1029/90RG02636, 1991.

Giorgi, F., Jones, C., and Asrar, G.: Addressing climate information needs at the regional level: the CORDEX framework, W.M.O. Buletin, 35  58, 175–183, 2009.

Gultepe, I. and Milbrandt, J. A.: Probabilistic Parameterizations of Visibility Using Observations of Rain Precipitation Rate, Relative Humidity, and Visibility, Journal of Applied Meteorology and Climatology, 49, 36–46, https://doi.org/10.1175/2009JAMC1927.1, https://DOI.org/10.1175/2009JAMC1927.1, 2010.





Hourdin, F., Musat, I., Bony, S., Braconnot, P., Codron, F., Dufresne, J.-L., Fairhead, L., Filiberti, M.-A., Friedlingstein, P., Grandpeix, J.-Y., Krinner, G., LeVan, P., Li, Z.-X., and Lott, F.: The LMDZ4 general circulation model: climate performance and sensitivity to parametrized physics with emphasis on tropical convection, Clim. Dyn., 27, 787–813, https://doi.org/10.1007/s00382-006-0158-0, http://dx.DOI.org/10.1007/s00382-006-0158-0, 2006.

Huang, H.-L., Yang, M.-J., and Sui, C.-H.: Water Budget and Precipitation Efficiency of Typhoon Morakot (2009), J. Atmos. Sci., 71, 112–129, https://doi.org/10.1175/JAS-D-13-053.1, 2014.

Jaeger, E. B. and Seneviratne, S. I.: Impact of soil moisture–atmosphere coupling on European climate extremes and trends in a regional climate model, Climate Dynamics, 36, 1919–1939, https://doi.org/10.1007/s00382-010-0780-8, https://doi.org/10.1007/s00382-010-0780-8, 2011.

Jourdier, B.: Ressource éolienne en France métropolitaine : méthodes d'évaluation du potentiel, variabilité et tendances, Climatologie: École Doctorale Polytechnique, 2015. Français. ph:+33 01238226, pp. 1–229, 2015.

Katragkou, E., García-Díez, M., Vautard, R., Sobolowski, S., Zanis, P., Alexandri, G., Cardoso, R. M., Colette, A., Fernández, J., Gobiet, A., Goergen, K., Karacostas, T., Knist, S., Mayer, S., Soares, P. M. M., Pytharoulis, I., Tegoulias, I., Tsikerdekis, A., and Jacob, D.: Regional climate hindcast simulations within EURO-CORDEX: evaluation of a WRF multi-physics ensemble, Geosci. Model Dev., 8, 603–618,
https://doi.org/10.5194/gmd-8-603-2015, https://www.geosci-model-dev.net/8/603/2015/, 2015.

Knist, S., Goergen, K., Buonomo, E., Christensen, O. B., Colette, A., Cardoso, R. M., Fealy, R., Fernández, J., García-Díez, M., Jacob, D., Kartsios, S., Katragkou, E., Keuler, K., Mayer, S., Meijgaard, E., Nikulin, G., Soares, P. M. M., Sobolowski, S., Szepszo, G., Teichmann, C., Vautard, R., Warrach-Sagi, K., Wulfmeyer, V., and Simmer, C.: Land-atmosphere coupling in EURO-CORDEX evaluation experiments, Journal of Geophysical Research: Atmospheres, 122, 79–103, https://doi.org/10.1002/2016JD025476, 2014.

Kotlarski, S., Keuler, K., Christensen, O. B., Colette, A., Déqué, M., Gobiet, A., Goergen, K., Jacob, D., Lüthi, D., van Meijgaard, E., Nikulin, G., Schär, C., Teichmann, C., Vautard, R., Warrach-Sagi, K., and Wulfmeyer, V.: Regional climate modeling on European scales: a joint standard evaluation of the EURO-CORDEX RCM ensemble, Geoscientific Model Development, 7, 1297–1333, https://doi.org/10.5194/gmd-7-1297-2014, 2017.

Kunkel, B. A.: Parameterization of Droplet Terminal Velocity and Extinction Coefficient in Fog Models, Journal of Climate and
Applied Meteorology, 23, 34–41, https://doi.org/10.1175/1520-0450(1984)023<0034:PODTVA>2.0.CO;2, https://DOI.org/10.1175/1520-0450(1984)023<0034:PODTVA>2.0.CO;2, 1984.

Manabe, S.: Climate and the ocean circulation, 1. the atmospheric circulation and the hydrology of the earth's surface, Mon. Weather Rev., 97, 739–774, https://doi.org/10.1175/1520-0493(1969)097<0739:CATOC>2.3.CO;2, 1969.

Mearns, L., Gutowski, W. . J., Jones, R., Leung, R., McGinnis, S., Nunes, A., and Qian, Y.: A regional climate change assessment program
for North America, EOS Transactions A.G.U., 90, 311–312, https://doi.org/10.1029/2009EO360002, 2009.

Milly, P. C. D.: Potential Evaporation and Soil Moisture in General Circulation Models, J. Climate, 5, 209–226, https://doi.org/10.1175/1520-0442(1992)005<0209:PEASMI>2.0.CO;2, https://doi.org/10.1175/1520-0442(1992)005<0209:PEASMI>2.0.CO;2, 1992.

Nakanishi, M. and Niino, H.: An Improved Mellor–Yamada Level-3 Model: Its Numerical Stability and Application to a Regional Prediction of Advection Fog, Bound.-Lay. Meteorol., 119, 397–407, https://doi.org/10.1007/s10546-005-9030-8, http://dx.doi.org/10.1007/s10546-005-9030-8, 2006.
s10546-005-9030-8, 2006.

Nielsen-Gammon, J. W., Powell, C. L., Mahoney, M. J., Angevine, W. M., Senff, C., White, A., Berkowitz, C., Doran, C., and Knupp, K.: Multisensor Estimation of Mixing Heights over a Coastal City, Journal of Applied Meteorology and Climatology, 47, 27–43, https://doi.org/10.1175/2007JAMC1503.1, https://DOI.org/10.1175/2007JAMC1503.1, 2008.



Nikulin, G., Jones, C., Giorgi, F., Asrar, G., Büchner, M., Cerezo-Mota, R., Christensen, O. B., Déqué, M., Fernández, J., Hänsler, A., van Meijgaard, E., Samuelsson, P., Sylla, M. B., and Sushama, L.: Precipitation Climatology in an Ensemble of CORDEX-Africa Regional Climate Simulations, Journal of Climate, 25, 6057–6078, https://doi.org/10.1175/JCLI-D-11-00375.1, https://DOI.org/10.1175/JCLI-D-11-00375.1, 2012.

5  Price, C. and Rind, D.: A simple lightning parameterization for calculating global lightning distributions, J. Geophys. Res.: Atmospheres, 97, 9919–9933, https://doi.org/10.1029/92JD00719, https://agupubs.onlinelibrary.wiley.com/doi/abs/10.1029/92JD00719, 1992.

Ruti, P. M., Somot, S., Giorgi, F., Dubois, C., Flaounas, E., Obermann, A., Dell'Aquila, A., Pisacane, G., Harzallah, A., Lombardi, E., Ahrens, B., Akhtar, N., Alias, A., Arsouze, T., Aznar, R., Bastin, S., Bartholy, J., Béranger, K., Beuvier, J., Bouffies-Cloché, S., Brauch, J., Cabos, W., Calmanti, S., Calvet, J.-C., Carillo, A., Conte, D., Coppola, E., Djurdjevic, V., Drobinski, P., Elizalde-Arellano, A., Gaertner,
10 M., Galàn, P., Gallardo, C., Gualdi, S., Goncalves, M., Jorba, O., Jordà, G., L'Heveder, B., Lebeaupin-Brossier, C., Li, L., Liguori, G., Lionello, P., Maciàs, D., Nabat, P., Önol, B., Raikovic, B., Ramage, K., Sevault, F., Sannino, G., Struglia, M. V., Sanna, A., Torma, C., and Vervatis, V.: Med-CORDEX Initiative for Mediterranean Climate Studies, Bulletin of the American Meteorological Society, 97, 1187–1208, https://doi.org/10.1175/BAMS-D-14-00176.1, https://DOI.org/10.1175/BAMS-D-14-00176.1, 2016.

Skamarock, W. C., Klemp, J. B., Dudhia, J., Gill, D. O., Duda, D. M. B. M. G., Huang, X.-Y., Wang, W., and Powers, J. G.: A Description
of the Advanced Research WRF Version 3, NCAR TECHNICAL NOTE, 475, NCAR/TN475+STR, 2008.

Smirnova, T. G., Benjamin, S. G., and Brown, J. M.: Case study verification of RUC/MAPS fog and visibility forecasts, Preprints, 9 th Conference on Aviation, Range, and Aerospace Meteorlogy, AMS, Orlando, FL, Sep. 2000, 2.3, 6, 2000.

Stackpole, J. and Cooley, D. S.: Revised method of 1000 mb height computations in the PE model, Technical Procedures Bulletin. U.S. Dept. of Commerce, National Oceanic and Atmospheric Administration, National Weather Service, 57, 6, 1970.

Vautard, R., Gobiet, A., Jacob, D., Belda, M., Colette, A., Déqué, M., Fernández, J., García-Díez, M., Goergen, K., Güttler, I., Halenka, T., Karacostas, T., Katragkou, E., Keuler, K., Kotlarski, S., Mayer, S., van Meijgaard, E., Nikulin, G., Patarčić, M., Scinocca, J., Sobolowski, S., Suklitsch, M., Teichmann, C., Warrach-Sagi, K., Wulfmeyer, V., and Yiou, P.: The simulation of European heat waves from an ensemble of regional climate models within the EURO-CORDEX project, Clim. Dyn., 41, 2555–2575, https://doi.org/10.1007/s00382-013-1714-z, https://DOI.org/10.1007/s00382-013-1714-z, 2013.

WMO: Manual on the Global Observing System, WMO, 544, 2010a.

WMO: Guide to Meteorological Instruments and Methods of Observation, Weather - Climate - Weather, pp. 1–176, 2010b.

Wong, J., Barth, M. C., and Noone, D.: Evaluating a lightning parameterization based on cloud-top height for mesoscale numerical model simulations, Geosci. Model Dev., 6, 429–443, https://doi.org/10.5194/gmd-6-429-2013, https://www.geosci-model-dev.net/6/429/2013/, 2013.

Yang, M. J., Braun, S. A., and Chen, D.-S.: Water Budget of Typhoon Nari (2001), Mon. Wather Rev., 139, 3809–3828, https://doi.org/10.1175/MWR-D-10-05090.1, 2011.

Yesad, K.: FULL-POS in the Cycle 41T1 of ARPEGE/IFS, Tech. Rep., Meteo-France, https://www.umr-cnrm.fr/gmapdoc/IMG/pdf/ykfpos43.pdf, 45, 2015.