# Peer review of "CORDEX-WRF v1.3: Development of a module for the Weather Research and Forecasting (WRF) model to support the CORDEX community"

_Geoscientific Model Development, 2018_

## Referee Comment (RC1) · Anonymous Referee #1 · 11 Dec 2018

This work introduces a tool that adds diagnostics to the WRF model while it is running in order to generate outputs of specific interest to CORDEX collaborators. As such it is valuable documentation of what is being added, and should be published for this information content.

General points

1. The technical description looks accurate to someone familiar with the model structure and methods.

2. WRF has a previous set of output diagnostics (wrfxtrm output option) that also probably includes necessary outputs for CORDEX. Is it true or not that those fields such

as daily max/min/mean surface values would also be required, or is the set presented here a complete requirement for CORDEX. The context of the existing wrfxtrm output needs to be mentioned.

Specific Points

1. p12, line 20. The model geopotential height is at full levels while others are at half levels. Does CORDEX expect that vertical staggering?

2. p13, line 10. The way these summations over time are done would preclude using adaptive time steps and just using a fixed step. This should be mentioned. In fact, wrong results could be obtained if adaptive steps are used. Slight modifications to the algorithms would allow for time-varying dt.

3. p14, line 7. It should be mentioned that sund has units of seconds. It was not obvious why it had such large values and a reader might first expect it to be in hours, for example.

4. p18, line 2. Is zp then used to simply vertically interpolate the wind? The description misses out this step.

5. p19, line 23. Why would not the Brasseur method also apply to gusts at 100 m? Assuming zp is above 100 m, it may be the same gusts.

6. Figure 13 and others. I am not sure the best figure quality has been achieved. The resolution looks low.

7. p26. Radiative fluxes. Are not the outward longwave and shortwave at the top also required? These would also be avaialable from other WRF fields with CAM and RRTMG options.

Minor Typos/Spelling

The paper could benefit from a editing read through, as I probably only caught a small percentage of errors.

[Figure]

1. p17, line 13. zclout

2. p37, line 25. doamins

3. p39, line 19. oder

4. p40, line 25. beneficed -> benefit

5. p41, line 9. cmorzization (cmorization?)
* * *

---

## Referee Comment (RC2) · Anonymous Referee #2 · 14 Dec 2018

**CORDEX-WRF v1.3: Development of a module for the Weather Research and Forecasting (WRF) model to support the CORDEX community**
*Lluís et al.*

**Summary**

The authors described the addition of CORDEX variables to the WRF model. Something that is truly needed in the community and will go a long way to help with reproducibility of the diagnostics for CORDEX as well as improving time to science for CORDEX participants.
I recommend acceptance of the publication after a major revision.

**Code development suggestions**

I would also recommend the authors contact the WRF development team and see about inclusion of this module in the standard release of the code. It will make it easier for new CORDEX participants and make this available to the broader WRF community.

It might be worth exploring *Runtime IO* as implemented by WRF. This will enable users to pick which variables to include and which to exclude, allowing for more flexibility.

**Major Comments**

One of my main concerns with the paper is that it will only be understood by users with a deep knowledge of the WRF code. I have a very deep understanding of the WRF code and still found the manuscript hard to follow. If this is acceptable, then it is fine, but some cleaning up will improve the readability of the manuscript.

The authors often refer to the levels of data required by CORDEX (Core Tier1 Tier2). It will be very helpful if a table is included with all the required variables in the different levels and indicate which are included in the new module and which compiler flag activates them.

The authors also go on to say that some additional post-processed is needed. What exactly is still needed and how should a user go about getting it done. I fully appreciate that not all components are provided, just some expansion on what is needed and left to the CORDEX participant will made this much clearer.

On page 6 there is a long list of variables which are available with the various compiler options. This shorthand list is not very helpful. Again, a table that the authors can point too will be better. If variables need to be called out, long names are going to make this much easier to read.

In a number of places, the authors say users need to make code changes. Is there a guide available so users know what to do? If yes, refer to it. Similarly, is it left to user to add new bits to the namelist, or are there examples? Please mention.

The authors provide a list of namelist options, but it is unclear if some of them are preferred/required by CORDEX. Please either say clearly that CORDEX does not have a preference and leaves it up to the research to pick which diagnostic method to use, or indicate which method is needed by CORDEX.

The term "generic" is introduced on page 11. I would like some explanation of what 'generic' means here. It becomes clear later, but will be helpful to have some explanation here.
Also, if the authors have a table as mentioned above, it can be used to indicate which diagnostics are scheme dependent and to which schemes.

Explain eta levels somewhere. Not all readers will be familiarly with the term and concept.

The manuscript can definitely benefit from an English speaker to review it for readability.

**Minor Comments**
Figure 1 is not very useful.

The word "specie" is used in a number of places. The correct word is "species". Specie refers to money.

Page 3 line 1: add an explanation of what netCDF is.

Some of the major English errors I picked up. There can be more.
Page 3, line 3
Page 5, line 4
Page 8, line 28
Page 9, lines 7 and 11
Page 11, line 16
Page 16, line 3
Page 24, line 25
Page 30, line 5
Page 37, line 8
Page 41, line 26

---

## Author Comment (AC1) · 25 Jan 2019

Author's answers to referee 1 of GMDD Discussion paper Geosci. Model Dev. Discuss., 2018 *"Development of a module for theWeatherResearch and Forecasting (WRF) model to support the CORDEX community"*, by Lluís Fita et al.

We appreciate the positive commentaries of the referee which certainly improve the quality of the manuscript.

**General points**

1. : The technical description looks accurate to someone familiar with the model structure and methods.

   *We got a similar comment from second referee. Certain parts of the document have been re-written in order to facilitate the comprehension of WRF structure to the non-familiar readers. Two new subsections has been added ('WRF code main characteristics' and 'Module implementation') and text has been re-organized accordingly.*

2. : WRF has a previous set of output diagnostics (wrfxtrm output option) that also probably includes necessary outputs for CORDEX. Is it true or not that those fields such as daily max/min/mean surface values would also be required, or is the set presented here a complete requirement for CORDEX. The context of the existing wrfxtrm output needs to be mentioned.

   *A new paragraph into the introduction has been added in order to contextualize the module:*
   *"This new module comes to complement the modifications introduced in the CLimate WRF (clWRF, http://www.meteo.unican.es/wiki/cordexwrf/SoftwareTools/ClWrf;, Fita et al. 2010). In clWRF climate statistical values (such as minimum, maximum and mean values) of certain surface variables where introduced into the model. At the same time, evolution of Green House Gases (GHG; $CO_2, N_2O, CH_4, CFC-11, CFC-12$) can be selected from an ASCII file instead of being hard coded. Before these modifications, WRF users could only retrieve those statistical values via post-processing the standard output of the model (at a certain frequency). With the clWRF modifications (incorporated into the WRF source code since version 3.5) statistical values are directly computed during model integration. This new CORDEX module proposes one step further by incorporating a series of new variables and diagnostics which are important for climate studies and currently WRF users can only obtain via post-processing the standard model output. At the same time, additional variables have been added into the WRF capabilities of output at pressure levels. At the current version of the module, if WRF adaptive time-step is used, some diagnostics with certain relation with the length of the time-step (e.g.: pr, prls, sund) will not properly work because module it is not yet adapted to it."*

**Specific Points**

1. `p12, line 20`: The model geopotential height is at full levels while others are at half levels. Does CORDEX expect that vertical staggering?

*As far we know, CORDEX does not expect any vertical staggering. In fact, CORDEX expects it at pressure levels which is already provided. In the module variable was already provided de-staggered, it is now clarified in the text as follows:*

*"As in the case of air-pressure, WRF model also integrates the perturbation of the geopotential field from a reference or base one. Thus to obtain the full geopotential height on staggered model $\eta$ levels, it is required to combine the two WRF fields and it is also de-staggered as it is shown in equation 1,*

$$
\begin{aligned}
zg_{staggered} &= PH + PHB \\
zg(k) &= 0.5\left(zg_{staggered}(k) + zg_{staggered}(k+1)\right), \ \ k = [1, dz]
\end{aligned}
\tag{1}
$$

*where PHB: WRF base geopotential height $(m^2 s^{-2})$, PH: WRF perturbation of the geopotential height $(m^2 s^{-2})$, $zg_{staggered}$: staggered geopotential height $k = [1, dimz + 1]$, zg: unstaggered geopotential height $k = [1, dimz]$"*

2. **p13, `line 10`:** The way these summations over time are done would preclude using adaptive time steps and just using a fixed step. This should be mentioned. In fact, wrong results could be obtained if adaptive steps are used. Slight modifications to the algorithms would allow for time-varying dt.

> *We agree on the comment. However we can not effort to take into account the required modification at this stage of the development of the module. Definitively it must be incorporated in a new update. A sentence has been added at the end of the paragraph and at the introduction in order to clarify this point.*
> *"Current version of the accumulations does not take into account configurations of the model with adaptive time-step. When adaptive time-step is used, we strongly discourage the use of these variables."*

3. **p14, `line 7`:** It should be mentioned that sund has units of seconds. It was not obvious why it had such large values and a reader might first expect it to be in hours, for example.

> *We followed CORDEX requiremnts which demands the variable in seconds. A sentence has been added to clarify it:*
> *"... implemented following equation 12 and provided in seconds."*

4. **p18, `line 2`:** Is zp then used to simply vertically interpolate the wind? The description misses out this step.

> *Not exactly, $z_p$ corresponds to the height until which the relation of equation 15 is satisfied. A new description of the variable has been added as follows:*
> *"$z_p$ height of the considered parcel (m, maximum height which satisfies equation 15)"*

5. **p19, `line 23`:** Why would not the Brasseur method also apply to gusts at 100 m? Assuming zp is above 100 m, it may be the same gusts.

> *We are not specialists in the methodology. As far we understood it, this might be certain for the wind gusts which are deflected from above 100 m. On doing that, wind gusts at surface and 100 m will be the same, since this will correspond to the deflected winds passing in their way until they reach surface. Without being confident with this assumption, we prefer to keep our diagnostic of the maximum wind at 100 m as a complement. A paragraph has been added in order to clarify this point as follows:*
> *"The calculation of wind gust at 100 m should follow a similar implementation used for calculating the `wsgsmax`, but at 100 m. An extrapolation of such turbulent phenomena would require a complete new set of equations which have not been placed yet. However, it could be considered as first approach to take the same wind gust as the one at the surface (when it is deflected from above 100 m). The assumption would be that the wind gust at 100 m would correspond to the deflected wind on its 'way' to the surface. Instead, as a way to complement, the estimation of maximum wind speed at 100 m is provided."*

6. `Figure 13 and others`: I am not sure the best figure quality has been achieved. The resolution looks low.

   *Figures have been re-rendered to higher PNG quality and now look at finer resolution*

7. `p26.`: Radiative fluxes. Are not the outward longwave and shortwave at the top also required? These would also be available from other WRF fields with CAM and RRTMG options.

   *They are already available from the module as 'RSUT' and 'RLUT' for short and long wave fluxes. They were not mentioned in the text of the article. They are now included as follows: "Outgoing radiative fluxes at top the atmosphere are also provided being 'rsut' for mean Top of the Atmosphere (TOA) outgoing shortwave radiation (in $Wm^{-2}$) and 'rlut' for longwave. However there is not a 'generic' implementation of these variables."*

**Minor Typos/Spelling**

1. : The paper could benefit from a editing read through, as I probably only caught a small percentage of errors.

   *A re-lecture of the entire manuscript has been carried out, and significant parts of the text have been re-written and improved.*

2. `p17, line 13`: zclout

   *Corrected to 'zcloud'*

3. `p37, line 25`: doamins

   *Corrected to 'domains'*

4. `p39, line 19`: oder

   *Corrected to 'other'*

5. `p40, line 25`: beneficed -> benefit

   *Correction done*

6. `p41, line 9`: cmorzization (cmorization?)

   *Being a new term there is not 'academic' source, thus googling it 'cmorization' has more entries, then changing to it*

---

## Author Comment (AC2) · 25 Jan 2019

Author's answers to referee 2 of GMDD Discussion paper Geosci. Model Dev. Discuss., 2018 *"Development of a module for theWeatherResearch and Forecasting (WRF) model to support the CORDEX community"*, by Lluís Fita et al.

We appreciate the positive commentaries of the referee which certainly improve the quality of the manuscript.

**Code development suggestions**

1. : I would also recommend the authors contact the WRF development team and see about inclusion of this module in the standard release of the code. It will make it easier for new CORDEX participants and make this available to the broader WRF community.

> *As porposed by the referee, we have already make contact with the WRF development team. We are currently working directly on the repository of the model with a new branch implementing the module in a dedicated branch. It might be possible that the module, after discussion of the developing team, might be included in the main source of the code. A new paragraph has been added in the conclusions.*
> *"The module is currently being implemented in the repository of the code in a dedicated branch. Once the module is fully implemented in the latest version of the model, and some additional tests are made, it might be possible that model developing team decides to include it in the main source of the code."*

2. : It might be worth exploring Runtime IO as implemented by WRF. This will enable users to pick which variables to include and which to exclude, allowing for more flexibility.

> *We agree with the referee that the 'Runtime IO' adds even more flexibility to the execution of the model. But in order to make it work, first, all variables have tobe declared and propertly incorporated into the code. Now WRF users can use the Runtime IO option to even increase their production by using this option with the CORDEX module. However, as our tests show, performance is not only a matter of which variables are written into file, but also which variables are passed and defined into the modules and subroutines.*

**Major Comments**

1. : One of my main concerns with the paper is that it will only be understood by users with a deep knowledge of the WRF code. I have a very deep understanding of the WRF code and still found the manuscript hard to follow. If this is acceptable, then it is fine, but some cleaning up will improve the readability of the manuscript.

> *We got a similar comment from first referee. Certain parts of the document have been re-written in order to facilitate the comprehension of WRF structure to the non-familiar readers. Two new subsections has been added ('WRF code main characteristics' and 'Module implementation') and text has been re-organized accordingly.*

2. : The authors often refer to the levels of data required by CORDEX (Core Tier1 Tier2). It will be very helpful if a table is included with all the required variables in the different levels and indicate which are included in the new module and which compiler flag activates them.

> *We agree with the referee, thus a new section labelled 'Requested CORDEX variables' in the appendix has been added with the requested information with the following details:*
> *"Here is provided a generic list of requested variables by CORDEX. Reader is advised that there is not a single CORDEX requirement variables list. It might depend on the experiment. However, hoping to provide a generic list of variables, a table with the CORDEX requirements in tables B1 to B3 is provided. The source of the table is from the ESGF servers at https: //www.earthsystemcog.org/doc/detail/1065/. Same variable might appear at different levels (Core, Tier-1, Tier-2) as function of the requested frequency and/or if should be provided as statistical value between output frequency or instantaneous value, as well as, depending on the experiment (FPS_Alps experiment requested additional variables provided in table B4)."*

3. : The authors also go on to say that some additional post-processed is needed. What exactly is still needed and how should a user go about getting it done. I fully appreciate that not all components are provided, just some expansion on what is needed and left to the CORDEX participant will made this much clearer.

> *We agree with the referee and a new paragraph has been added into the conclusions:*
> *"The module provides almost all the required all the CORDEX variables. However, user still needs to perform some post-processing of the output data in order to meet CORDEX standards. Mainly:*
>
> - *Computation of the required different statistical values as daily, monthly and seasonal extremes (minimum, maximum, accumulations, means)*
>
> - *Cmorization of the output understood as: 1 file per variable, right metadata and attributes and general CF-compilant standard specifications"*

4. `On page 6`: there is a long list of variables which are available with the various compiler options. This shorthand list is not very helpful. Again, a table that the authors can point too will be better. If variables need to be called out, long names are going to make this much easier to read.

> *The idea with the list of variables was to provide a summarized overview of the variables provided with the module. In the appendix, a more detailed description of the variables is provided in table form. A reference to the section now is included into the text as follows:*
> *"According to the value given to the pre-compilation CDXWRF flag, different amount of variables is written out to the 'wrfcdx' output file (see more detail in appendix C):"*

5. : In a number of places, the authors say users need to make code changes. Is there a guide available so users know what to do? If yes, refer to it. Similarly, is it left to user to add new bits to the namelist, or are there examples? Please mention.

> *In order to reduce the text of the article, we preferred to avoid to include all the details in the text. We already make reference to a WIKI page with more details and explanations. At he same time there is a README provided with the module with deeper details on the manage, use and compilation of the module. We included in the article a larger explanation in:*
> *• at the 'The CORDEX module' section:*
> *"Here we introduce the module and we explain the modifications introduced in the model. The steps necessary to follow in order to compile and use the module are provided as well. For a complete and detailed description of the steps to follow, reader is referred to the wiki page of the module:* `http://wiki.cima.fcen.uba.ar/mediawiki/index.php/CDXWRF` *and the README file provided with the module labeled* `README.cordex`*. The module has been implemented following standards of modularity which facilitates the upgrading and the introduction of new variables to it. "*
> *• at the 'Module use' section as follows:*
> *"A series of steps have to be made in order to use the CORDEX-WRF modifications. These steps encompass compilation of the module and its specific set-up to be used during the execution time of the model and are described in the following subsections."*

6. : The authors provide a list of namelist options, but it is unclear if some of them are preferred/required by CORDEX. Please either say clearly that CORDEX does not have a preference and leaves it up to the research to pick which diagnostic method to use, or indicate which method is needed by CORDEX.

> *We agree with the referee, thus two specifications has been added into the table for the different values of the namelist. Accordingly new text has been added to the caption of the table as follows:*
> *"The methodologies preferred by CORDEX are marked by [a], the ones without preference by CORDEX are marked by [b] in these cases, users can select the method according to their experience."*

7. : The term "generic" is introduced on page 11. I would like some explanation of what 'generic' means here. It becomes clear later, but will be helpful to have some explanation here. Also, if the authors have a table as mentioned above, it can be used to indicate which diagnostics are scheme dependent and to which schemes.

> *The term 'generic' was already introduced at page 3 a the paragraph starting at line 31 of the manuscript as follows:*
> *"The modifications also aim to establish a series of homogenization of certain diagnostics. These diagnostics can be computed following different methodologies, and consequently they may be model and/or even physical parameterization dependent. In order to avoid dependency on the model configuration (mainly sensitivity to the choice of the available different physical schemes), and to allow for a fair comparison between different simulations, a series of additional 'generic' definitions of some diagnostics are presented when possible."*
> *Also a mark has been added to the table of CORDEX variables when they present a dependency on the scheme being used.*

8. : Explain eta levels somewhere. Not all readers will be familiarly with the term and concept.

> *An explanation about the levels has been added into the 'WRF code main characteristics' section as follows:*
> *"WRF model integrates the atmosphere using $\eta$ as vertical variable (see more detail in, Skamarock et al., 2008) defined in equation 1 (being $p_{surf}$: surface pressure, $p_{top}$: pressure at top, $p$ hydrostatic pressure and $\eta = 1$, surface and $\eta = 0$ on top of the atmosphere)."*

$$\eta = \frac{p - p_{top}}{p_{surf} - p_{top}} \tag{1}$$

9. : The manuscript can definitely benefit from an English speaker to review it for readability.

> *A re-lecture of the entire manuscript has been carried out, and significant parts of the text have been re-written and improved.*

**Minor Comments**

1. `Figure 1`: is not very useful.

> *Figure is used in different parts of the text and it is used to provide detail about when the variables are initialized/computed. We would like to keep it as a graphical way to help the explanations*

2. : The word "specie" is used in a number of places. The correct word is "species". Specie refers to money.

> *Corrected as suggested*

3. `Page 3 line 1`: add an explanation of what netCDF is.

> *A short explanation has been added as follows:*
> *"... criteria in netcdf format (netCDF, Network Common Data Form https://www.unidata.ucar.edu/software/netcdf/ a binary self-describing and machine-independent file format)."*

Some of the major English errors I picked up. There can be more.

4. `Page 3, line 3`:

> *Sentence has been rewritten to:*
> *"... exist which facilitate the manipulation of netcdf files ..."*

5. `Page 5, line 4:` Apart

   *Changed to 'Aside' in all text*

6. `Page 8, line 28:`

   *Sentence has been rewritten:*
   *"statistics: values obtained as statistics of consecutive instantaneous values along a given period of time"*

7. `Page 9, lines 7 and 11:`

   *Sentences have been re-written as:*
   *"... require other variables or requires some of them at different frequency of output in comparison to a standard CORDEX requested list of variables ..."*
   *"... the diagnostic is updated following the configuration from the namelist ..."*

8. `Page 11, line 16:`

   *Sentence has been re-written:*
   *"These are the basic variables required by CORDEX"*

9. `Page 16, line 3:`

   *Sentence has been re-written:*
   *"...when the spatial smoothing is applied"*

10. `Page 24, line 25:`

    *Sentence has been re-written to:*
    *"In order to solve this problem, ..."*

11. `Page 30, line 5:`

    *Second 'are' has been removed*

12. `Page 37, line 8:`

    *Sentence has been rewritten as:*
    *"Usually these experiments require long periods of time for the period of simulation."*

13. `Page 41, line 26:`

    *Sentence has been re-written to:*
    *"This would facilitate the creation of a community of ..."*

---

## Author Response (AR2)

Author's answers to the topical editor of GMD paper Geosci. Model Dev., 2019 *"Development of a module for the Weather Research and Forecasting (WRF) model to support the CORDEX community", by Lluís Fita et al.*

*We appreciate the positive commentaries of the topical editor which certainly improve the quality of the manuscript.*

**General points**

1. `Abstract, at line 10`*: Where you state the problem at hand, you also refer to final homogenization. But, as you have now very clearly explained in the paper, the new code does not ensure final homogenization. If a reader reads the abstract alone, I think they will assume this is also done. Hence I recommend to add just one more sentence to the abstract to explain what the code does not ensure final homogenization for CORDEX.*

   *As suggestion of the editor, we added the following last sentence intot the abstract:*
   *'The module performs neither additional statistics over different periods of time nor homogenization of the output data.'*

2. `Paragraphing - Section 2.1`*: I suggest to make into 1 paragraph, rather than 2. Page 10, line 15, you have a single sentence paragraph here? it really breaks the flow. Please revise the entire manuscript and check for these. Also page 20, line 16, I assume you forget to change lines here?*

   *These minor paragraphing errors have been corrected as suggested*

3. `Section 2.1`*: It is great that there is now a general intro to WRF. However, you should also refer the reader to the WRF technical doc, and users guide in this section for more details.*

   *The suggested references have been added as follows:*
   *'For further technical details of the model, reader is referred to WRF technical note (http: // www2. mmm. ucar. edu/ wrf/ users/ docs/ arw_ v3. pdf , Skamarock et al. 2008) and the users' guide web (http: // www2. mmm. ucar. edu/ wrf/ users/ docs/ user_ guide_ v4/ contents. html ).'*

4. `General English`*: Please avoid using phrases such as "it is required", rather use "the user is required to". Same thing on page 9, line 26, "this will require to perform" should be "this will require the user to perform". Please check the entire manuscript again for similar minor issues. also page 41, line 8 "it should be tested", what is "it", just say "this new module should be tested". Also, page 43, "can be overcame" should be "can be overcome". While copy-editing will pick up these minor mistakes, it is also your responsibility to ensure there is a minimum of such mistakes. Please review carefully again.*

   *These English mistakes have been corrected. A completed and detailed read of the article has been done and some minor additional errors have been fixed.*